



# Numerical Investigation of Regenerative Wind Farms Featuring Enhanced Vertical Energy Entrainment

YuanTso Li[1], Wei Yu[1], Andrea Sciacchitano[1], and Carlos Ferreira[1]

[1]Delft University of Technology, Faculty of Aerospace Engineering, Kluyerweg 1, 2629 HS Delft, the Netherlands

**Correspondence:** YuanTso Li (Y.Li-18@tudelft.nl)

**Abstract.** Numerical simulations of wind farms consisting of innovative wind energy harvesting systems are conducted. The novel wind harvesting system is designed to generate strong lift (vertical force) with lifting-devices. It is demonstrated that the tip-vortices generated by these lifting-devices can substantially enhance wake recovery rates by altering the vertical entrainment process. Specifically, the wake recovery of the novel systems is based on vertical advection processes instead of turbulent mixing. Additionally, the novel wind energy harvesting systems are hypothesized to be feasible without requiring significant technological advancements, as they could be implemented as Multi-Rotor Systems with Lifting-devices (MRSLs), where the lifting-devices consist of large airfoil structures. Wind farms with these novel wind harvesting systems, namely MRSLs, are termed *regenerative wind farm*, inspired by the concept that the upstream MRSLs actively entrain energy for the downstream ones. With the concept of regenerative wind farming, much higher wind farm capacity factors are anticipated. Specifically, the results indicate that the wind farm efficiencies can be nearly doubled by replacing traditional wind turbines with MRSLs under the tested conditions, and this disruptive advancement can potentially lead to a profound reduction in the cost of future renewable energy.

## 1 Introduction

In the wind energy industry, wind turbines are often arranged in clusters, leveraging closer spacings for economic and operational benefits (Meyers and Meneveau, 2012; Sørensen and Larsen, 2021). These clusters are known as wind farms. However, densely packed wind turbines result in Annual Energy Production (AEP) losses due to the turbine-turbine wake interactions. The more tightly packed the turbines, the more pronounced the negative impact on AEP (Stevens et al., 2016). These losses are substantial, with reported AEP reductions ranging from 10 to 25% for large-scale offshore wind farms such as Horns Rev I & Nysted (Barthelmie et al., 2009, 2010). Moreover, predictions indicate that AEP losses due to wakes could reach more than 60% for wind farms on a very large scale (infinite wind farm) with spacings similar to the typical ones (e.g., $7D$ in streamwise and $5D$ lateral directions, where $D$ is the rotor diameter) (Dupont et al., 2018; Calaf et al., 2010). Note that the above-mentioned considerations are for conventional wind farms that consist of three-bladed Horizontal-Axis Wind Turbines (HAWTs), the prevailing concept in today's commercial wind farms (Manwell et al., 2010).

The AEP drop mentioned in the previous paragraph is attributed to the fact that the kinetic energy carried by the incoming wind is depleted by upstream turbines, and the energy replenishing rates cannot sustain the downstream turbines to extract as





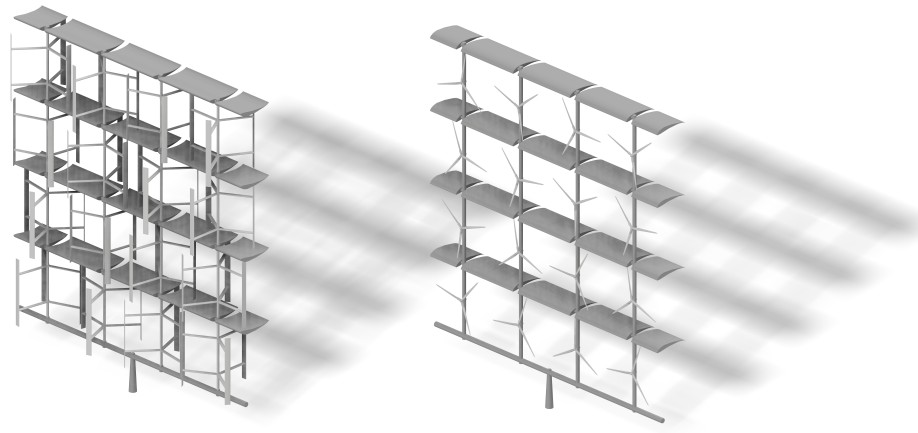

**Figure 1.** Conceptualizing the proposed design of the innovative wind energy harvesting systems, namely multi-rotors systems with lifting-devices (MRSL). Left: MRSL consisting of Vertical Axis Wind Turbines (VAWT) with airfoils/wings that lift the wake upward. Right: MRSL consists of Horizontal Axis Wind Turbines (HAWT) with airfoils/wings that lift the wake downward.

much energy as those in the first row of a wind farm (Porté-Agel et al., 2020). Note that the energy is mainly replenished by entraining from above the wind farms. This is due to the fact that wind farms are built close to the ground or sea surface, and they extend in both streamwise and lateral directions. However, without significant mean vertical flow in conventional wind farms, the primary source of vertical energy (momentum) entrainment is through the turbulent mixing process, relying on the

Reynolds stress terms (Calaf et al., 2010; VerHulst and Meneveau, 2015). Typical rates of vertical energy entrainment are about 1 to 2 W/m$^2$ for conventional wind farms with HAWTs (Dupont et al., 2018; VerHulst and Meneveau, 2015) (estimated based on infinite wind farms with conventional spacings, e.g., the ranges of streamwise and lateral spacings are around $7D$ to $5D$), which is significantly lower than typical installation capacities (e.g., $\sim 7$ W/m$^2$ for a wind farm with $7D$ and $5D$ streamwise and lateral spacings, freestream wind speed being 10 m/s, and power coefficient of the turbines being $0.54$) (Barthelmie et al.,

2009; Bosch et al., 2019). This indicates that the efficiencies of large wind farms with conventional designs are limited by the low vertical entrainment rates.

To overcome the aforementioned limitations of the current conventional wind farms, we adopt the strategy of introducing lifting-devices onto the wind energy harvesting systems. These lifting-devices can induce strong vertical flows, leading to a significant vertical advection process and thus enhancing vertical energy entrainment. To the authors' best knowledge, this con-

cept was first studied by Bader et al. (2018), where they carried out numerical analysis of HAWTs coupled with lifting-devices close to their rotors in various configurations. Their promising results showed that the power performance of the downstream turbines was substantially improved with the implementation of the lifting-devices. However, they did not propose a way to install the lifting-devices, as they were suspended without support in the computational domain. Very recently, Broertjes et al. (2024) and Martins et al. (2024) have also studied this concept using both experimental and numerical methods, and these stud-

ies were based on the idea proposed by Ferreira et al. (2024). Unlike Bader et al. (2018), an innovative design, the Multi-Rotor





System coupled with Lifting-devices (MRSL), was proposed. The system comprises several sub-rotors, each in the form of VAWT or HAWT (Vertical/Horizontal Axis Wind Turbine). The proposed design, illustrated in Figure 1, highlights the designated positions to mount the lifting-devices, where the airfoils/wings themselves serve as structural components. Their results showed that, due to the strong vertical flow induced by the lifting-devices, the wake recovery rate of MRSL can reach more than $90\%$ at a distance of $5D$ downstream (based on available power, which is $\propto u^3$), whereas a typical HAWT achieves less than $40\%$ at a similar distance (Li et al., 2024a). This yeilds a significant enhancement in wake recovery. Additionally, it should be noted that although the concept of MRSL came out very recently, the implementation of this design may not require major technological breakthroughs, as the technology for multi-rotor systems already exists (Jamieson and Branney, 2012; Watson et al., 2019).

Building on the work of Broertjes et al. (2024) and Martins et al. (2024), this study further investigated the aerodynamics of wind farms consisting of MRSLs using numerical method. These wind farms are termed *regenerative wind farms* by Ferreira et al. (2024). The name reflects the idea that upstream MRSLs actively entrain energy for the downstream ones. At this point, it is suggested that the proposed MRSLs and the concept of regenerative wind farms could be a groundbreaking concept for the wind energy industry. This concept has the potential to revolutionize wind energy by fundamentally altering the process of vertical energy entrainment. Unlike conventional wind farms, regenerative wind farms replenish flow energy vertically through the mean components of the flow rather than relying on Reynolds stress terms, which is likely to significantly elevate their wind farm efficiency. If successfully implemented, this approach promises not only significant economic advantages but also a reduction in the space required to generate the same power output compared to conventional wind farms. Achieving these goals could enhance the benefits of wind energy while minimizing its environmental and spatial impacts, marking a transformative advancement in renewable energy. To validate the groundbreaking potential of MRSLs in transforming the vertical entrainment process, this study conducts a comprehensive numerical analysis of regenerative wind farms, setting the stage for a significant leap forward in wind farm efficiency.

## 2 Working principles and specifications of multi-rotor system with lifting-devices

### 2.1 Working principles of MRSLs

As mentioned in the previous section, one of the key reasons that conventional wind farms suffer from slow wake recovery rates is the absence of vertical flow. Regenerative wind farms counter this shortcoming by introducing vertical advection through the placement of lifting-devices onto MRSLs. How this concept works is depicted by the vertical velocity fields $w$ inside regenerative wind farms presented in Figure 2, where the active exchange of flow between the upper and lower layers can be observed. This concept is inspired by the flow field induced by a wing described by the classic lifting-line theory (Anderson, 2011). As depicted in Figure 3, the vorticity/circulation system of a wing can be simplified as a horseshoe vortex. The horseshoe vortex consists of two tip vortices and a bound vortex. Due to the induction field of this vortex system, particularly from its tip vortices, the induced flow $\boldsymbol{u}_i$ behind the wing has a non-zero vertical component (perpendicular to both the freestream and spanwise directions), resulting in $w_i \neq 0$. Additionally, both the strength and direction of $w_i$ are affected by the wing's



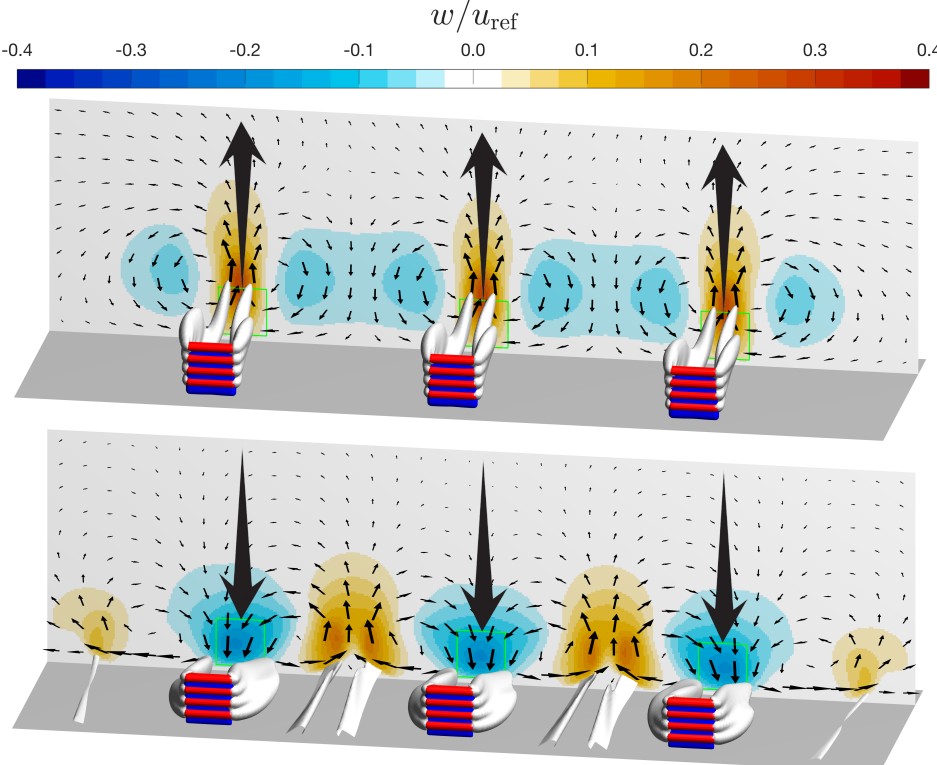

**Figure 2.** Iso-surfaces of the magnitude of streamwise vorticity ($|\omega_x|$, silver) together with the contour plots of the vertical velocity fields $w$. The arrows on the contours depicts the direction of the in-plane velocity, and note that the lengths of the arrows are scaled by the square root of the in-plane velocity's norms. MRSLs are represented with red and blue surfaces, where blue surfaces are thrusting devices while the red surfaces are the lifting-devices, respectively. The plots are based on the solutions of cases **Upward-lifting** (top) and **Downward-lifting** (bottom) in Table 3. The MRSLs depicted are the ones at the $4^{\text{th}}$ row, and the contours are plotted at $x/D = 22.0$.

configuration. The strength of $w_i$ is governed by the lift per span of the wing, with a higher lift generating a stronger circulation
$\Gamma$ and thus a larger $w_i$, as explained by the Kutta-Joukowski theorem and Helmholtz's theorem (Anderson, 2011). The direction
of $w_i$ can be altered by flipping the wing, that is, swapping the locations of the pressure side and the suction side. Moreover,
stacking multiple wings vertically can further amplify $w_i$. Thus, in this work, MRSLs are equipped with several wings, referred
to as the lifting-devices, to increase the magnitude of $w_i$. By arranging these lifting-devices as shown in Figure 1, the flow at
different altitudes behind MRSLs are exchanged vertically due to the non-zero $w_i$. It is this non-zero vertical flow induced by
the lifting-devices that fundamentally changes the mechanism of vertical energy entrainment within regenerative wind farms
(Ferreira et al., 2024).

Based on the configurations of the lifting-devices/wings of MRSLs, the lift exerted by MRSLs can both be upward or
downward. In this work, the configuration that exerts upward-lift onto the flow is termed **Upward-lifting** while the one that
exerts downward-lift is termed **Downward-lifting**. With the contours of vertical velocity $w$ together with the iso-surfaces of





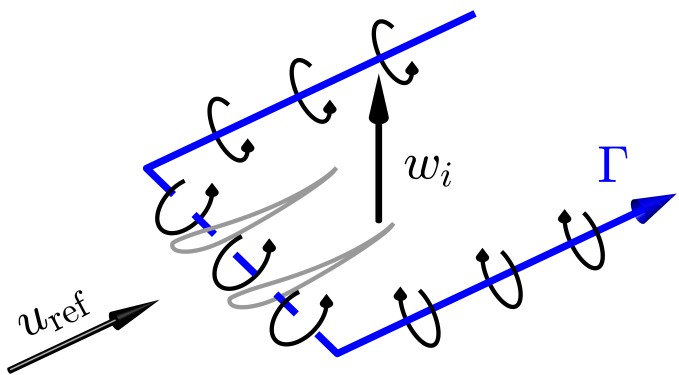

**Figure 3.** A simplified sketch of the vortex/circulation system of a wing is presented. The vortex system is depicted as a horseshoe vortex, indicated by the blue line with angles in the sketch, with the direction of the circulation $\Gamma$ shown by an arrow. The solid lines in the horseshoe vortex represent the trailing vortices, while the dashed line represents the bound vortex. Note that, with the orientation of the wing in this figure, the vertical component of the induced flow $w_i$ right behind the wing is upward.

the streamwise vorticity magnitudes $|\omega_x|$ in Figure 2, it can be seen that the flow at lower altitudes is channeled upward while the flow at higher altitudes is brought downward for both **Upward-lifting** and **Downward-lifting**. Note that $\omega_x$ represents the tip-vortices. The visual representation in Figure 2 demonstrates the presence of the trailing vortices consistently guides the flow in the lower layers upward and bring the flow in the upper layers downward, enhancing the vertical exchange process. The working principle of the lifting-devices here is akin to the vortex generators on the wings of modern aircraft and the blades of

contemporary wind turbines, but on the scale of wind farms, which is much larger (Ferreira et al., 2024).

## 2.2 Specifications of MRSLs

In this work, the shape of the frontal area of MRSL is set as a square (as shown in Figure 1) with a side length $D$ of 300 m, where the height of the rotor center $z_{\mathrm{rc}}$ is 186 m, corresponds to a clearance of 36±m. The lifting-devices of MRSL consist of four straight wings without any twist. These wings are placed at 100%, 75%, 50%, and 25% of the MRSL's height as depicted

in Figure 4. Table 1 lists the key parameters of the MRSL used in this work. Note that MRSL in Figure 4 degenerates from Figure 1, where the sub-rotors are represented with an actuator disk (blue surface) and the lifting-devices/wings are represented with four actuator lines (red surfaces). This simplification enables more efficient numerical modeling (Mikkelsen, 2004), and the detailed parametrization is provided in Section 3.4.

The thrust force exerted by an MRSL is calculated based on the sampled local velocity, where the thrust coefficient $C_T$ is

set to 0.7. According to classic actuator disk theory (Manwell et al., 2010), $C_T = 0.7$ gives a power coefficient $C_P$ of 0.54 (see Section 3.4 for more explanations). Note that $C_P = 0.54$ is around the design values for modern large scale wind turbines (Bak et al., 2013; Gaertner et al., 2020). The lifting-devices of an MRSL consist of four straight wings with constant profile, constant twist angle, constant chord length $c$, and a span of $D$. The chord length of the wings is set to $c = D/8$, and the airfoil used





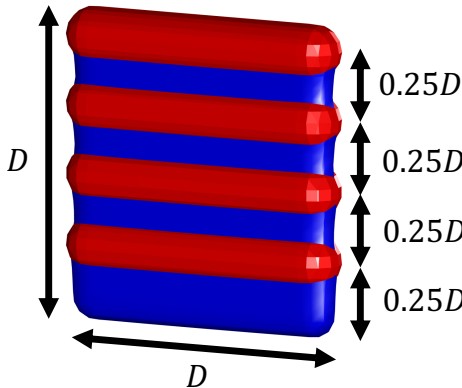

**Figure 4.** MRSL represented with an actuator disk (blue) and four actuator lines (red). Note that the MRSL showed here is a degenerated form of Figure 1.

**Table 1.** Specifications of MRSL modelled in the current work. $D$, $c$, and $z_{\mathrm{rc}}$ are MRSL's side length, chord length, and height of the rotor center, respectively. Designed $C_P$ (power coefficient) is estimated based on classic actuator disk theory theory (Manwell et al., 2010) with $C_T = 0.7$. Designed $T^R$ and $P^R$ are the designed thrust and power of an entire MRSL estimated based on $C_T = 0.7$ and $C_P = 0.54$ with $u_{\mathrm{ref}} = 10$ m/s.

| Parameter | Value |
|---|---|
| $D$ | 300 m |
| $c$ | 37.5 m |
| Wing span | 300 m |
| Airfoil shape | S1223 (Selig and Guglielmo, 1997) |
| $z_{\mathrm{rc}}$ | 186 m |
| $C_T$ | 0.70 |
| Designed $C_P$ | 0.54 |
| Designed $T^R$ | 3,858 kN |
| Designed $P^R$ | 29.85 MW |

is S1223 airfoil (Selig and Guglielmo, 1997). The airfoil coordinate and the lift-drag polar (calculated with the chord-based
Reynolds number $Re_c$ being $2 \times 10^7$ using XFOIL version 6.99 (Drela, 1989)) for S1223 airfoil are plotted in Figure 5.




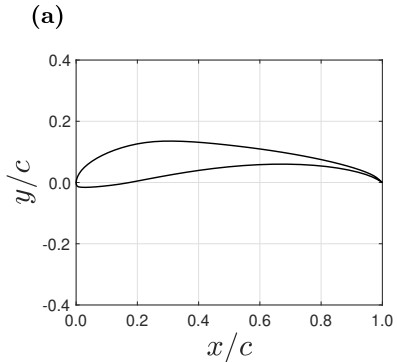
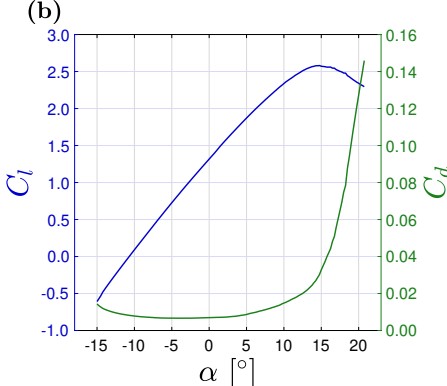

**Figure 5.** (a): XY-plot of a the cross-section of airfoil S1223 (Selig and Guglielmo, 1997). (b): The lift/drag polar of airfoil S1223 obtained with XFOIL version 6.99 (Drela, 1989) with $Re_c = 2 \times 10^7$.

## 3 Methodology

### 3.1 Numerical setup and computational domain

Numerical simulations of this work are conducted with *OpenFOAM v2106* (OpenCFD Ltd., 2021), an open-source finite-volume-based CFD solver. The flow is treated as incompressible and Newtonian ($\rho = 1.225$ kg/m$^3$ and $\nu = 1.5 \times 10^{-5}$ m$^2$/s),

and neither thermal effects nor Coriolis force are considered. Reynolds-averaged Navier-Stokes (RANS) approach is employed. While higher fidelity models such as Large-Eddy Simulation (LES) are available for wind energy applications, RANS is selected for its lower computational demands (Thé and Yu, 2017), making it more suitable for rapid testing of new concepts and allowing for a broader parametric study. For the turbulence closure, $k$-$\omega$ SST model (Menter, 1994) is chosen as it is the most widely used turbulence model in wind energy applications (Thé and Yu, 2017). A sensitivity test on the turbulence model

is conducted, showing that the choice of turbulence model has little impact on the conclusions drawn from this work (see A). In this work, all model coefficients of the $k$-$\omega$ SST model are set to the default values provided by OpenFOAM v2106 (OpenCFD Ltd., 2021) (e.g., $\beta^* = 0.09$).

The key governing equations of RANS with $k$-$\omega$ SST model are written in Equations 1 to 3, which are the equations for continuity, transport of momentum, and transport of modelled turbulence kinetic energy (denoted as TKE or $k$). In these

equations, $u_i$, $p$, $k$, $\omega$, $\rho$, $\nu$, $f_{\text{body},i}$, $S_{ij}$, $\tau_{ij}$, and $\nu_T$ denote the $i$th component of velocity, static pressure, turbulence kinetic energy, turbulence specific dissipation, fluid density, kinematic (molecular) viscosity, $i$th component of the body forces applied on the flow, shear strain tensor, Reynolds stress tensor, and eddy viscosity. Note that all quantities just mentioned are time-averaged. The definition of $S_{ij}$ and the modeling of $\tau_{ij}$ are written in Equation 4. For brevity, certain equations related to the $k$-$\omega$ SST model, such as the transport equation of $\omega$ and the calculation of $\nu_T$, have been omitted. The readers are referred to

the OpenFOAM v2106 documentation (OpenCFD Ltd., 2021) for further details.





$$\frac{\partial u_i}{\partial x_i} = 0 \tag{1}$$

$$u_j \frac{\partial u_i}{\partial x_j} = -\frac{1}{\rho}\frac{\partial p}{\partial x_i} + \frac{\partial}{\partial x_j}\Big(2\nu S_{ij} + \tau_{ij}\Big) + \frac{f_{\text{body},i}}{\rho} \tag{2}$$

$$u_j \frac{\partial k}{\partial x_j} = \tau_{ij}\frac{\partial u_i}{\partial x_j} - \beta^* \omega k + \frac{\partial}{\partial x_j}\left[\Big(\nu + \nu_T\Big)\frac{\partial k}{\partial x_j}\right] \tag{3}$$

$$S_{ij} \stackrel{\Delta}{=} \frac{1}{2}\left(\frac{\partial u_i}{\partial x_j} + \frac{\partial u_j}{\partial x_i}\right), \qquad \tau_{ij} = 2\nu_T S_{ij} - \frac{2}{3}k\delta_{ij} \tag{4}$$

The spatial discretization schemes used are linear-upwind (`Gauss linearUpwind`) for divergence and second-order central differencing (`Gauss linear` with limiter) for gradient and Laplacian. Pressure-velocity system is solved using SIMPLE (Semi-Implicit Method for Pressure Linked Equations) algorithm.

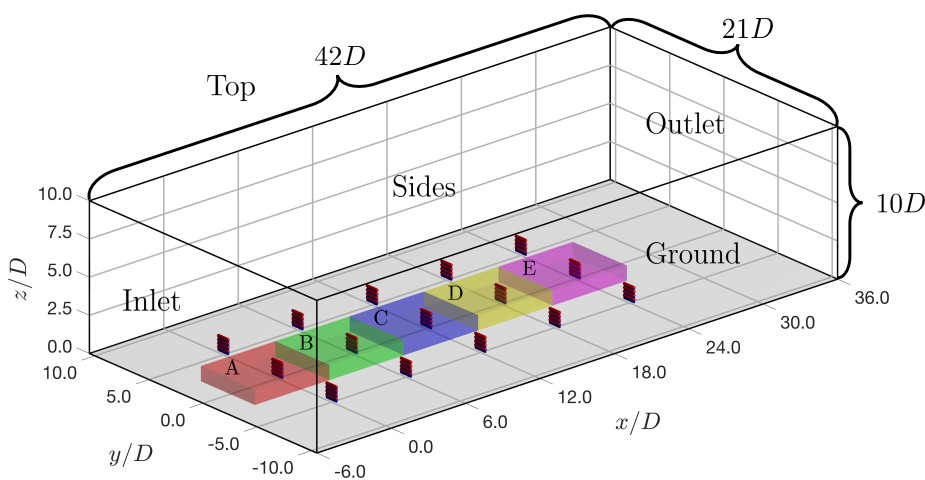

**Figure 6.** Diagram depicting the computational domain and the layout of regenerative wind farm. The inflow comes from the bottom-left to the top-right. The wind farm consists of fifteen multi-rotor systems with lifting-devices (MRSL), having a layout of five rows and three columns. The deep blue surfaces represent the rotor part of MRSLs (thrusting devices), while the deep red surfaces indicate the lifting-devices of MRSL. The semi-transparent volumes annotated with alphabet are the control volumes used in Figure 15.

## 3.2   Computational domain and boundary conditions

The computational domain for the simulations are illustrated in Figure 6. A Cartesian coordinate system is employed, with pos-
itive $x$ pointing downstream and positive $z$ pointing upward. The mesh is generated using application `blockMesh`, consisting



of uniformly sized cubic cells with a grid size of $\Delta = D/25$ in all three directions. The dimensions of the computational domain are $42D \times 21D \times 10D$ in the $x$, $y$, and $z$ (streamwise, lateral, vertical) directions, respectively, comprising approximately 137.8M cells. Additionally, a grid independence test is carried out in B, confirming that a grid size of $\Delta = D/25$ is adequate for this study.

**Table 2.** Boundary conditions used for the simulations cases of regenerative wind farms immersed in ABL. `atmBoundaryLayer` and `WallFunction` are abbreviated to `ABL` and `WF`. For instance, `ABLInletVelocity` stands for `atmBoundaryLayerInletVelocity`.

|  | Inlet | Outlet | Ground | Top | Sides |
|---|---|---|---|---|---|
| $\boldsymbol{u}$ | ABLInletVelocity | inletOutlet | noSlip | inletOutlet | inletOutlet |
| $p$ | zeroGradient | uniformFixedValue | zeroGradient | zeroGradient | zeroGradient |
| $k$ | ABLInletK | inletOutlet | kqRWF | inletOutlet | zeroGradient |
| $\omega$ | ABLInletOmega | inletOutlet | omegaWF | zeroGradient | zeroGradient |
| $\nu_T$ | calculated | calculated | atmNutkWF | calculated | calculated |

A built-in library of OpenFOAM v2106, `atmosphericModels` (Richards and Hoxey, 1993; Hargreaves and Wright, 2007), is used to model the atmospheric boundary layer (ABL). The inlet profiles for the (mean) streamwise velocity $u$ and turbulence kinetic energy $k$ are given in Equations 5 and 6, respectively (see Figure 12 for the generated freestream profile). $z_0$ is the surface roughness length, which is set to $10^{-4}$ m, a typical value for offshore environment (Manwell et al., 2010). $u_{\text{ref}}$ are the reference velocity at the height of the rotor center $z_{\text{rc}}$, which is set to 10 m/s. $C_1$ and $C_2$ are the two constants that are set to $0.81$ and $1.0$ in order to make the turbulence intensity TI $= 8\%$ at $z = z_{\text{rc}}$, where TI is defined as $(2k/3)^{0.5}/|\boldsymbol{u}|$. This value corresponds to the typical turbulence intensity for the offshore environment (Hansen et al., 2012). The boundary conditions used in this work are listed in Table 2. For more detailed specifications of the used boundary conditions, readers are referred to the OpenFOAM v2106 documentation (OpenCFD Ltd., 2021). Additionally, note that any constants and model coefficients not explicitly mentioned are set to their default values. e.g., $\kappa = 0.41$ and $C_\mu = 0.09$.

$$u = \frac{u^*}{\kappa} \ln\left( \frac{z + z_0}{z_0} \right), \qquad u^* = \frac{u_{\text{ref}} \kappa}{\ln\left( \frac{z_{\text{rc}} + z_0}{z_0} \right)} \tag{5}$$

$$k = \frac{(u^*)^2}{\sqrt{C_\mu}} \sqrt{C_1 \ln\left( \frac{z + z_0}{z_0} \right) + C_2} \tag{6}$$

### 3.3 Wind farm layout

All the simulations in this work share the same wind farm layout, which consists of five rows and three columns. MRSLs in each column are fully aligned with the direction of the freestream. The mid-column of the wind farm is placed at the centerline of the computational domain, and the $1^{\text{st}}$ row is located $6D$ from the inlet. The lateral distance between any two columns





is $5D$, and the streamwise distance between the rows is $6D$. The origin of the coordinate system is set at the $1^{\text{st}}$ row of the mid-column, as indicated in Figure 6.

### 3.4 Modeling multi-rotor system with lifting-devices

The multi-rotor systems with lifting-devices (MRSL) introduced previously are parameterized using a square actuator "disk"
(called disk for historical reasons) together with four actuator lines, as mentioned in Section 2. With actuator techniques, the effects of MRSL geometry are replaced by body force fields (term $\boldsymbol{f}_{\text{body}}$ in Equation 2). This allows avoiding the exceptionally high computational cost required to resolve the boundary layer around the complex geometry (Mikkelsen, 2004; Sorensen and Shen, 2002). These actuator methods are realized in OpenFOAM using a customized library building upon `actuationDiskSource` (a built-in library of OpenFOAM v2106) and `turbinesFoam` (Bachant et al., 2019), and we
term it `flyingActuationDiskSource` (Li et al., 2024b).

The rotors of MRSL (thrusting devices) are modeled with 25 by 25 actuator elements situated on the same streamwise plane. These actuator elements have the same inter-distance in both lateral and vertical directions. The rotors of each MRSL have non-uniform loading based on the velocities sampled dynamically at each actuator element. $T^{\text{ele}}$ and $C_T^{\text{ele}}$ in Equation 7 denote the thrust force exerted by the actuator element and the corresponding thrust coefficient, respectively. $u_{\text{in}}^{\text{ele}}$ is the undisturbed
inflow velocity seen by the actuator element. For all simulation cases in this work, the element-based area $A^{\text{ele}}$ is $D^2/625$. The $C_T^{\text{ele}}$ targeted for each element for all MRSL is set to $0.70$. However, because the undisturbed inflow velocity perceived by an actuator element ($u_{\text{in}}^{\text{ele}}$) can vary when simulating wind farms and there is no universal method to define where to measure $u_{\text{in}}^{\text{ele}}$, estimating the value of $T^{\text{ele}}$ for an actuator element directly based on $C_T^{\text{ele}}$ using Equation 7 is challenging. To overcome this challenge, $T^{\text{ele}}$ of this work is estimated based on the locally sampled velocity $u_{\text{ls}}^{\text{ele}}$ and the corrected thrust coefficient
$C_T^{*,\text{ele}}$ as expressed in Equation 8. Note that $u_{\text{ls}}^{\text{ele}}$ is the velocity sampled exactly at where the actuator element situated. Unlike $u_{\text{in}}^{\text{ele}}$, position of $u_{\text{in}}^{\text{ls}}$ does not have ambiguity. $C_T^{*,\text{ele}}$ and $C_T^{\text{ele}}$ are linked through the classic actuator disk (one-dimensional momentum) theory (Manwell et al., 2010), which stated $C_T^{\text{ele}}$ can be expressed as Equation 9 based on the axial induction factor $a^{\text{ele}}$. After dividing/rearranging Equations 7 and 8 and applying the classic actuator disk theory (Equation 9), expression of $C_T^{*,\text{ele}}$ is obtained with $C_T^{\text{ele}}$ and $a^{\text{ele}}$ as written in Equation 10. This method had been successfully implemented by Calaf et al.
(Calaf et al., 2010). Through Equation 9, it can be calculated that $C_T^{\text{ele}} = 0.7$ infers $a^{\text{ele}} = 0.23$, which leads to $C_T^{*,\text{ele}} = 1.17$.

$$T^{\text{ele}} = 0.5\rho\,(u_{\text{in}}^{\text{ele}})^2 A^{\text{ele}} C_T^{\text{ele}} \tag{7}$$

$$T^{\text{ele}} = 0.5\rho\,(u_{\text{ls}}^{\text{ele}})^2 A^{\text{ele}} C_T^{*,\text{ele}} \tag{8}$$

$$C_T^{\text{ele}} \simeq 4a^{\text{ele}}(1 - a^{\text{ele}}), \qquad a^{\text{ele}} \triangleq 1 - \frac{u_{\text{ls}}^{\text{ele}}}{u_{\text{in}}^{\text{ele}}} \tag{9}$$



$$C_T^{*,\text{ele}} = C_T^{\text{ele}} \left( \frac{u_{\text{in}}^{\text{ele}}}{u_{\text{ls}}^{\text{ele}}} \right)^2 \simeq \frac{C_T^{\text{ele}}}{(1 - a^{\text{ele}})^2} \tag{10}$$

After obtaining the value of $T^{\text{ele}}$ through Equation 8, the force is projected onto the CFD grid with Equation 11, where $\boldsymbol{f}^{\text{ele}}$ is the force vector exerted by the actuator element and $\boldsymbol{f}_{\text{body}}^{\text{ele}}(\boldsymbol{x})$ is the body force field on the CFD grid projected by $\boldsymbol{f}^{\text{ele}}$ at position $\boldsymbol{x}$. $\boldsymbol{\xi}^{\text{ele}}$ denotes the position vector of the actuator element. The projection is done by the Gaussian normalization kernel, it is introduced to improve the robustness of the numerical modeling (Mikkelsen, 2004; Sorensen and Shen, 2002), where $\varepsilon$ is called smearing factor. For the actuator elements of MRSL's rotors, $\boldsymbol{f}^{\text{ele}} = -T^{\text{ele}} \, \hat{\boldsymbol{e}}_x$ is assigned and its smearing factor, denoted as $\varepsilon^R$, is set to $1.0 \, \Delta$ as it is commonly used for actuator disk (Mikkelsen, 2004; Wu and Porté-Agel, 2011). The thrust and power of the rotor ($T^R$ and $P^R$) are calculated after projecting the body force fields using Equation 12. Here, $i$ and $j$ represent the indices for positions and actuator elements, respectively.

$$\boldsymbol{f}_{\text{body}}^{\text{ele}}(\boldsymbol{x}) = \boldsymbol{f}^{\text{ele}} \, \eta_\varepsilon(\|\boldsymbol{x} - \boldsymbol{\xi}^{\text{ele}}\|), \qquad \eta_\varepsilon(d) = \frac{1}{\varepsilon^3 \pi^{3/2}} \exp\left[ -\left( \frac{d}{\varepsilon} \right)^2 \right] \tag{11}$$

$$T^R = \sum_i \sum_j f_{j,\text{body}}^{\text{ele}}(\boldsymbol{x_i}) \, \Delta^3, \quad P^R = -\sum_i \sum_j u_{i,\text{ls}}^{\text{ele}} f_{j,\text{body}}^{\text{ele}}(\boldsymbol{x_i}) \, \Delta^3 \tag{12}$$

As mentioned in Section 2, the lifting-devices of MRSLs are parameterized with four actuator lines, with each having 25 equally-spaced actuator elements lining up in the lateral direction, and these actuator elements are in the same plane as those of the rotors. The forces to project are calculated based on the blade element approach, where $\boldsymbol{f}^{\text{AL}}$ is calculated based on the velocity sampled and the airfoil polar as written in Equation 13. $\boldsymbol{u}^{\text{AL}}$ is the flow velocity for an actuator element of an actuator line. $\boldsymbol{f}_l^{\text{AL}}$, $\boldsymbol{f}_d^{\text{AL}}$, $C_l$, and $C_d$ are the lift/drag forces and their corresponding coefficients. In this work, $C_l$ and $C_d$ are based on the polar data of the S1223 airfoil plotted in Figure 5. $\Delta^{\text{AL}}$ is the span length to which the actuator element corresponds. In this work, $\Delta^{\text{AL}} = D/25$. $\hat{\boldsymbol{e}}_s$, $\hat{\boldsymbol{e}}_l$, and $\hat{\boldsymbol{e}}_d$ are the unit vectors in the directions of spanwise, lift, and drag, respectively. Note that $\hat{\boldsymbol{e}}_s \| \pm \hat{\boldsymbol{e}}_y$ (depending on the lifting direction) and $\hat{\boldsymbol{e}}_l \| (\boldsymbol{u}^{\text{AL}} \times \hat{\boldsymbol{e}}_s)$. In this work, $\boldsymbol{u}^{\text{AL}}$ is obtained by averaging the 20 velocity samples sampled on a circular path with the actuator element at the center (line averaging). The sampling points are equidistant and the normal direction of the enclosed surface is parallel to the spanwise direction. The radius of the circle is set to $r^{\text{AL}} = 3\Delta \simeq c$. Single-point sampling is avoided to achieve better robustness (Melani et al., 2021). Note that since $\boldsymbol{u}^{\text{AL}} \perp \hat{\boldsymbol{e}}_l$ and the wings are stationary, the lift forces of the wings do not do any work on the flow.

$$\boldsymbol{f}^{\text{AL}} = \left( \boldsymbol{f}_l^{\text{AL}}, \boldsymbol{f}_d^{\text{AL}} \right) = 0.5 \rho \left( u^{\text{AL}} \right)^2 c \, \Delta^{\text{AL}} \left( C_l(\alpha)\hat{\boldsymbol{e}}_l, C_d(\alpha)\hat{\boldsymbol{e}}_d \right) = f_z^{\text{AL}} \, \hat{\boldsymbol{e}}_z + f_x^{\text{AL}} \, \hat{\boldsymbol{e}}_x \tag{13}$$

Gaussian normalization kernel (see Equation 11, where $\boldsymbol{f}^{\text{ele}}$ is replaced with $\boldsymbol{f}^{\text{AL}}$) is used again to project the forces of the lifting-devices on to the CFD grid. While for the smearing factor $\varepsilon$, instead of assigning a single value, the values of $\varepsilon$ for



the actuator lines (denoted as $\varepsilon^W$) are calculated based on the relative wing position as described in Equation 14 ($r/D = 0.0$ correspond to the middle of the wing). This approach was introduced by Jha et al. (Jha et al., 2014). Compared with the experimental results of the load of a finite wing, it has been shown that using this distribution of $\varepsilon^W$ outperformed the case using a single value for $\varepsilon^W$ (Jha et al., 2014; Jha and Schmitz, 2018). For the current work, $n_{\max}$ is assigned as 3.0, and the distribution of $\varepsilon^W$ along the wing used in this work is plotted in Figure 7.

$$\varepsilon^W = n_{\max} \Delta \sqrt{1 - \left(\frac{2r}{D}\right)^2}, \qquad -\frac{1}{2} \le \frac{r}{D} \le \frac{1}{2} \tag{14}$$

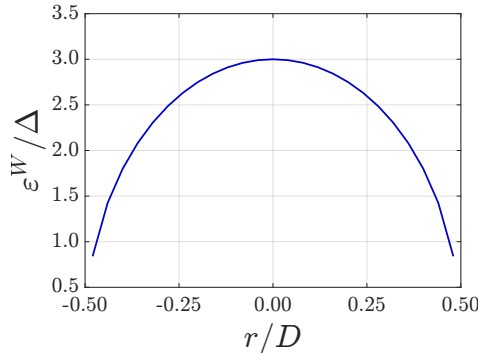

**Figure 7.** The distribution of $\varepsilon^W$ across the wings used in this work. $\Delta = D/25$ is the grid size of the mesh used in the cases listed in Table 3.

Using a similar method to obtain $T^R$ and $P^R$ (Equation 12), the total lift $L^W$ (vertical force) and the total induced drag $D_{\mathrm{ind}}^W$ (streamwise force) of the four wings of an MRSL are obtained through Equation 15. It should be noted that the directions of $\boldsymbol{L}^W$ and $\boldsymbol{D}_{\mathrm{ind}}^W$ are based on the global coordinate system, which is different from $\boldsymbol{f}_l^{\mathrm{AL}}$ and $\boldsymbol{f}_d^{\mathrm{AL}}$, where they are based on the airfoil coordinate system.

$$L^W = \sum_i \sum_j \boldsymbol{f}_{j,\mathrm{body}}^{\mathrm{AL}}(\boldsymbol{x_i}) \cdot \hat{\boldsymbol{e}}_z \, \Delta^3, \quad D_{\mathrm{ind}}^W = \sum_i \sum_j \boldsymbol{f}_{j,\mathrm{body}}^{\mathrm{AL}}(\boldsymbol{x_i}) \cdot \hat{\boldsymbol{e}}_x \, \Delta^3 \tag{15}$$

To adjust the magnitudes of the lift force exerted by the lifting-devices, the wings of MRSLs are pitched in the simulations by varying their pitch angles $\theta_p$. Specifically, $\theta_p$ of each wing is adjusted so that the angle of attack $\alpha$ at the midpoint of the wing yields a specified value for $C_{l,\mathrm{mid}}$ (the lift coefficient at the midpoint of the wing). Since the inflow conditions for each of the MRSLs' wings differ, $\theta_p$ varies for each wing. The adjustment of $\theta_p$ is programmed and carried out automatically during the simulations. Note that each wing is pitched as a whole and has a constant twist angle along its entire span. For a demonstration, see C, where profiles of $\alpha$ along the wings are presented. Additionally, for the MRSLs in the cases equipped with lifting-devices, all their wings are pitched to make $C_{l,\mathrm{mid}} = 2.5$, except for the cases in D.



### 3.5 Transport equations of flow energies

In addition to evaluating the performance of MRSLs based on their power outputs, analysis is also conducted using the terms

of the energy transport equations based on the control volume approach. This analysis aims to distinguish the primary source terms for wake recoveries.

We start with the two transport equations for MKE (mean kinetic energy, also denoted as $K$) and TKE (turbulence kinetic energy, also denoted as $k$) provided in Equations 16 and 17, where the physical meaning of each term is labeled. The definition of MKE is given in Equation 18. The transport equations for MKE (Equation 16) and TKE (Equation 17) are derived by

multiplying $\rho u_i$ and $\rho$ with Equations 2 and 3, respectively. Some rearrangements are then performed using the continuity equation (Equation 1) and the chain rule.

$$
\underbrace{\frac{\partial u_j K}{\partial x_j}}_{\text{advection of MKE}} = \underbrace{-\frac{\partial u_j p}{\partial x_j} + \rho \frac{\partial}{\partial x_j}\left[u_i\left(2\nu S_{ij} + \tau_{ij}\right)\right]}_{\text{work done by surface forces}}
$$

$$
\underbrace{-2\rho\nu S_{ij}\left(\frac{\partial u_i}{\partial x_j}\right)}_{\text{viscous dissipation}} \quad \underbrace{-\rho\tau_{ij}\left(\frac{\partial u_i}{\partial x_j}\right)}_{\text{transfer to TKE}} \quad \underbrace{+\rho u_i f_{\text{body},i}}_{\text{work done by body forces}} \tag{16}
$$

$$
\underbrace{\frac{\partial u_j \rho k}{\partial x_j}}_{\text{advection of }k} = \underbrace{\rho\tau_{ij}\left(\frac{\partial u_i}{\partial x_j}\right)}_{\text{transfer from MKE}} \quad \underbrace{-\rho\beta^*\omega k}_{\text{dissipation of }k} + \underbrace{\frac{\partial}{\partial x_j}\left[\rho\left(\nu + \nu_T\right)\frac{\partial k}{\partial x_j}\right]}_{\text{diffusion of }k} \tag{17}
$$

$$K \triangleq \frac{1}{2}\rho u_i u_i \tag{18}$$

The transport equation for the total energy (resolved plus modeled) in differential form can be obtained by adding the two energy equations (Equations 16 and 17). This equation is integrated over a control volume (CV) to examine the energy balance, resulting in Equation 19. The divergence theorem is applied, with CS denoting the control surface bounding the CV. It is worth noting that the term *MKE diffusion plus pressure work* essentially represents the work done by surface forces on the

control volumes. Due to the high Reynolds number in this study (e.g., $Re_c = u_{\text{ref}}\,c/\nu > 10^7$ and $Re_D = u_{\text{ref}}\,D/\nu > 10^8$), the primary contributor of *MKE diffusion plus pressure work* is the turbulent shear stress, which is modeled through the Reynolds shear stress $\tau_{ij}$. Additionally, the signs for each term on the left-hand side of the equation are rearranged so that positive values correspond to energy gains for a CV, and vice versa for the terms on the right-hand side. The viscous dissipation term is omitted because $\nu \ll \nu_T$ due to the high Reynolds number. Furthermore, a residuals term $\mathcal{R}$ is introduced to account for discrepancies,

including the viscous dissipation term, losses due to the parasitic drag of the wings, errors from discretization, interpolation errors, and other factors.





**Table 3.** Test matrix of the tested simulation cases.

| Case name | Direction of lift |
|---|---|
| **Without-lifting** | - |
| **Upward-lifting** | upward |
| **Downward-lifting** | downward |

$$\rho \oint_{\text{CS}} \left( \underbrace{-u_j K}_{\text{MKE advection}} + \underbrace{\left[ u_i\left(2\nu S_{ij} + \tau_{ij}\right) - \frac{u_j p}{\rho} \right]}_{\text{MKE diffusion plus pressure work}} \underbrace{-u_j k + \left[ \left(\nu + \nu_T\right)\frac{\partial k}{\partial x_j} \right]}_{\text{TKE advection plus diffusion}} \right) \mathrm{d}S_j$$

$$= \int_{\text{CV}} \left( \underbrace{-u_i\, f^R_{\text{body},i}}_{\text{Power extraction}} + \underbrace{\rho \beta^* \omega k}_{\text{TKE dissipation}} \right) \mathrm{d}V + \underbrace{\mathcal{R}}_{\text{Residuals}} \quad (19)$$

### 3.6 Test matrix

This study includes three simulations. The three cases are **Without-lifting**, **Upward-lifting**, and **Downward-lifting**, as listed in Table 3. In the case **Without-lifting**, MRSLs are not equipped with lifting-devices, serving as the reference case. In the case **Upward-lifting**, the lifting-devices on MRSLs exert upward lift, and one of the immediate effects is that the wakes right behind MRSLs are directed upward. Similarly, in the case **Downward-lifting**, the lifting-devices are designed to exert downward lift (orientation of the wings are flipped compared to the category **Upward-lifting**), sending the wakes right behind 265 MRSLs downward.

For the lifting-devices of the MRSLs in cases **Upward-lifting** and **Downward-lifting**, their wings are pitched during the simulations to make $C_{l,\text{mid}}$ for each wing being 2.5 ($C_{l,\text{mid}}$ is the lift coefficient at the mid-span of a wing, see the end of Section 3.4). Note that based on some rough estimations using the specifications provided in Table 1, $C_{l,\text{mid}} = 2.5$ allows an MRSL to generate a vertical force that is in similar magnitude to the thrust force of its rotors.

## 4 Results and Discussions

### 4.1 Forces exerted by MRSLs

The thrust of the MRSL's rotors together with the lift (vertical force) and the induced drag (streamwise force) of the MRSL's lifting-devices/wings are plotted in Figure 8. The three cases listed in Table 3 are displayed. For the loading profiles of the MRSL's wings, see C.

$\widehat{T}^R$, $\widehat{L}^W$, and $\widehat{D}^W_{\text{ind}}$ in Equation 20 and Figure 8 are the normalized thrust $T^R$, lift $L^W$, and induced drag $D^W_{\text{ind}}$ of MRSL (Equations 12 and 15), respectively. These forces are normalized against $T^R$ measured at the 1st-row-mid-column of the case





**Without-lifting** in Table 3, denoted as $T^R\big|^{\mathbf{WL}}_{1^{\text{st}},\text{mid}}$, which is 3.87 MN. This value is very close to the designed value of 3.86 MN, which is based on letting $C_T = 0.7$ and a reference velocity of $u_{\text{ref}} = 10$ m/s.

$$\widehat{T}^R \triangleq \frac{T^R}{T^R\big|^{\mathbf{WL}}_{1^{\text{st}},\text{mid}}}, \qquad \widehat{L}^W \triangleq \frac{\left|L^W\right|}{T^R\big|^{\mathbf{WL}}_{1^{\text{st}},\text{mid}}}, \qquad \widehat{D}^W_{\text{ind}} \triangleq \frac{\left|D^W_{\text{ind}}\right|}{T^R\big|^{\mathbf{WL}}_{1^{\text{st}},\text{mid}}} \tag{20}$$

Operator $< \cdot >$ in this work, including those in Figure 8, indicates row-averaging. For example, $< \widehat{T}^R >$ denotes the row-averaged normalized rotor thrust. Noted that the results show that the value differences between the middle and side columns are at most $1\%$ for $T^R$, $L^W$, and $D^W_{\text{ind}}$.

As shown in the left and middle panels of Figure 8, as designed, $< \widehat{L}^W >$ for the MRSLs in the two cases with lifting-devices are similar to their $< \widehat{T}^R >$, while the case **Without-lifting** has zero lift. Additionally, for both **Upward-lifting** and
**Downward-lifting**, it can be observed that their $< \widehat{T}^R >$ values are much higher than those of **Without-lifting** from the $2^{\text{nd}}$ row onward, despite the lifting-devices also introduce significant $< \widehat{D}^W_{\text{ind}} >$, as shown in the right panel of Figure 8. Specifically, it is found that the thrust for the two cases with lifting-devices only slightly decrease from the $1^{\text{st}}$ to the $2^{\text{nd}}$ row, with $< \widehat{T}^R >$ remaining above $80\%$, and the decreasing trend ceases from the $3^{\text{rd}}$ row onward. In contrast, for the case without lifting-devices, $< \widehat{T}^R >$ drop significantly from the $1^{\text{st}}$ to $2^{\text{nd}}$ row, falling below $60\%$, and continued to decrease row by row.
By the $3^{\text{rd}}$ row, $< \widehat{T}^R >$ for the two cases with lifting-devices are more than double compared to the case **Without-lifting**. Additionally, the fact that the forces for the MRSLs in cases **Upward-lifting** and **Downward-lifting** remain relatively stable from the $3^{\text{rd}}$ to the $5^{\text{th}}$ row suggests that these values would likely be sustainable if the regenerative wind farms had more rows. Furthermore, higher values of $< \widehat{T}^R >$ suggests that the streamwise velocity experienced by an MRSL at a given row is much higher when lifting-devices are equipped. This is further confirmed by the plots and contours presented in later sections
(Sections 4.3 and 4.3.3).

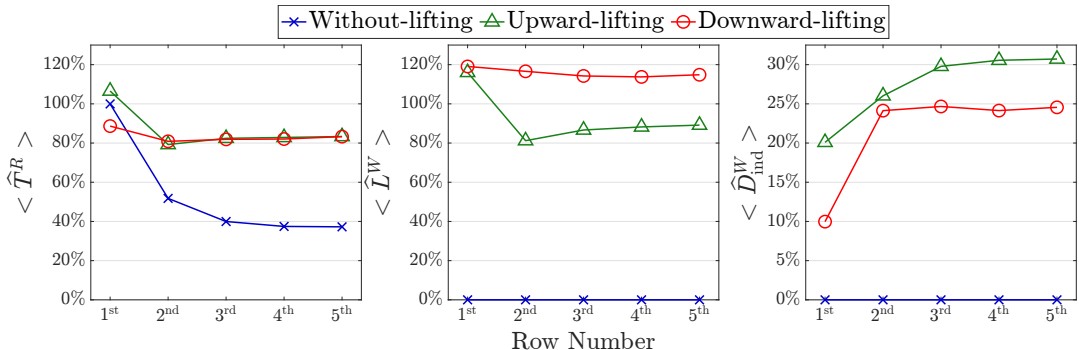

**Figure 8.** The normalized row-averaged thrust of MRSL's rotor ($\widehat{T}^R$, left) together with the vertical ($\widehat{L}^W$, middle) and streamwise ($\widehat{D}^W_{\text{ind}}$, right) force components of the MRSL's lifting-devices. The normalization is done by dividing the reference rotor thrust, which is based on the MRSL at $1^{\text{st}}$-row-mid-column of the case **Without-lifting**. The legends correspond to the case name introduced in Table 3.





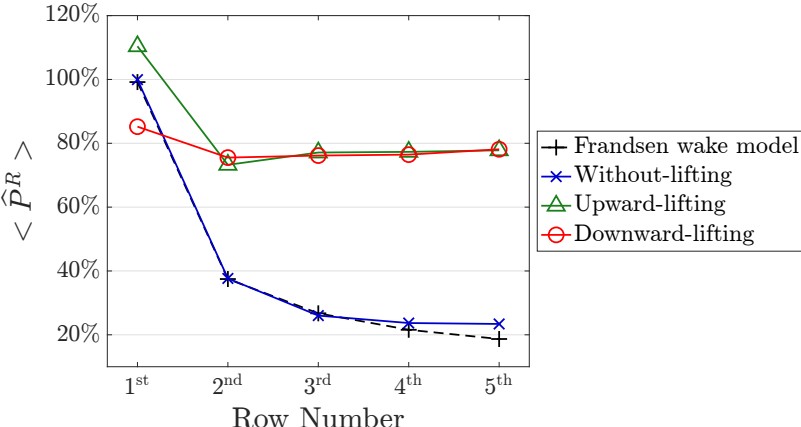

**Figure 9.** The normalized row-averaged rotor power of MRSL's rotor ($\widehat{P}^R$). The normalization is done by dividing the values by the rotor power of the MRSL situated at $1^{\text{st}}$-row-mid-column of the case **Without-lifting**. Calculation of the use of Frandsen wake model is detailed in E.

## 4.2 Power harvested by MRSL

Figure 9 presents the normalized row-averaged power $< \widehat{P}^R >$ harvested by the rotors of MRSLs for the three cases listed in Table 3. These values are plotted alongside those predicted by the Frandsen wake model (Frandsen et al., 2006) (see E). As in the previous subsection, the rotor power $P^R$ is normalized based on the MRSL located at the $1^{\text{st}}$-row-mid-column of the
case **Without-lifting**. The reference power, denoted as $P^R\big|_{1^{\text{st}},\text{mid}}^{\textbf{WL}}$, is 30.1 MW. This value corresponds to a power density of 11.1 W/m$^2$. This power density is calculated by dividing $P^R\big|_{1^{\text{st}},\text{mid}}^{\textbf{WL}}$ by the footprint area of an MRSL, which in this study is $6D \times 5D$.

A very good agreement was found between the CFD results for the case **Without-lifting** (case **N0_0**) and the predictions of the Frandsen wake model, which supports the validity of the numerical framework used in this work.

As expected, $\widehat{P}^R$ of the three cases are highly correlated with their $\widehat{T}^R$ (as indicated by Equation 12), with the cases having lifting-devices also exhibiting higher $\widehat{P}^R$. However, in terms of the magnitudes, the relative differences in $\widehat{P}^R$ between cases with and without lifting-devices are greater than those in $\widehat{T}^R$, since $\widehat{P}^R$ is proportional to the cube of the sampled velocity, while $\widehat{T}^R$ is proportional to the square of it (Equations 8 and 12).

Examining the values of $< \widehat{P}^R >$ for the $1^{\text{st}}$ row of the seven cases in Figure 9, it is observed that $< \widehat{P}^R >$ for the case
**Upward-lifting** is higher than that of the case **Without-lifting**. In contrast, the case **Downward-lifting** exhibits the opposite behavior. This can be attributed to the wings (lifting-devices) of the MRSLs acting as diffuser-like devices. A straightforward explanation is that the bound circulations of the wings (Anderson, 2011) either accelerate or decelerate the flow velocity crossing the rotor (thrusting devices) of an MRSL, depending on the configuration of the lifting-devices. Previous studies have reported similar phenomena with comparable configurations (Bader et al., 2018). Although this effect influences the power





**Table 4.** The relative power densities of the regenerative wind farms of the seven cases in Table 3 and the values predicted by the Frandsen wake model (see E). $100\%$ correspond to $11.1$ W/m$^2$, which is the power density of the MRSL at the $1^{\text{st}}$-row-mid-column in case **Without-lifting**.

| Case number | Relative power density[%] |
|---|---|
| **Frandsen wake model** | 40.8 |
| **Without-lifting** | 42.2 |
| **Upward-lifting** | 83.2 |
| **Downward-lifting** | 78.3 |

output of MRSLs, it is overshadowed by the effects of the enhanced wake recoveries due to the lifting-devices. Therefore, it is not discussed nor quantified in the rest of this work.

When comparing the power output row by row across the entire regenerative wind farm, it is found that the cases with lifting-devices have significantly higher values for $<\widehat{P}^R>$ compared to the case without at and after the $2^{\text{nd}}$ row. Specifically, $<\widehat{P}^R>$ for the cases **Upward-lifting** and **Downward-lifting** at the $2^{\text{nd}}$ row are more than double that of the case **Without-**

**lifting**. Remarkably, for the $3^{\text{rd}}$ to $5^{\text{th}}$ rows, $<\widehat{P}^R>$ for the two cases with lifting-devices are more than triple compared to that without. Furthermore, despite the relatively small spacing (around $5.3D_{\text{cir}}$, considering the shape effects of the rotor, see E), $<\widehat{P}^R>$ for the cases with lifting-devices remains at least $80\%$ of the reference power up to the $5^{\text{th}}$ row. This significantly outperforms conventional wind turbines (i.e., HAWT), which typically maintain around $40\%$ or $60\%$ when the inflow is aligned with the wind farm layout and when the streamwise spacing is $5D_{\text{cir}}$ or $7D_{\text{cir}}$, respectively (Barthelmie et al., 2010; Li et al.,

2024a; Wu and Porté-Agel, 2015). These power output results underscore the profound potential of the concept of regenerative wind farm, supporting the current proposal.

The overall performance of the regenerative wind farms is evaluated based on power density, which serves as a measure of the efficiency of the regenerative wind farms. Table 4 lists the relative power densities of the regenerative wind farms, with $100\%$ corresponding to $11.1$ W/m$^2$, which is the power density of the MRSL at the $1^{\text{st}}$-row-mid-column in case **Without-**

**lifting** mentioned earlier. Similarly, as has been seen in the plot of $<\widehat{P}^R>$ (Figure 9), the result of the case **Without-lifting** has very good agreement with the prediction given by the Frandsen wake model. By comparing the values in Table 4, it is evident that the two cases with lifting-devices (cases **Upward-lifting** and **Downward-lifting**) have power densities that are nearly double that of the case **Without-lifting**, increasing from approximately $40\%$ to around $80\%$. In other words, the power losses due to wake interactions among the regenerative wind farms are reduced from roughly $60\%$ to about $20\%$ by introducing

lifting-devices. These results demonstrate the capabilities of MRSLs and the tremendous potential of regenerative wind farms in achieving significantly higher wind farm efficiencies than conventional wind farms.





### 4.3 Flow fields characterization

#### 4.3.1 Three-dimensional flow structures

Figure 10 illustrates the three-dimensional flow structures of the simulated wind farms based on streamwise velocity. All three
cases in Table 3 are considered. The plots cover the mid-column of the regenerative wind farms, with the positions of the
MRSLs represented by deep-blue surfaces for the rotors and deep-red surfaces for the lifting-devices/wings. The low-speed
wakes are depicted by light-blue iso-surfaces, corresponding to $u/u_{\mathrm{ref}} = 0.65$. Additionally, several $x$-planes color-coded by
streamwise velocity $u$ are displayed, with the directions of in-plane velocity indicated by arrows.

In the plot for the case **Without-lifting**, it is evident that the MRSLs after the 2$^{\mathrm{nd}}$ row are generally immersed in the wakes
of the upstream ones, resulting in significantly lower inflow velocities compared to the 1$^{\mathrm{st}}$ row. Additionally, based on the
arrows in the plot, it can be seen that vertical velocity are generally absent, making its wake recovery rates slow. Consequently,
as shown in Figure 9, the power outputs of the MRSLs after the 2$^{\mathrm{nd}}$ row are much lower compared to those in the 1$^{\mathrm{st}}$ row for
the case **Without-lifting**.

In the case **Upward-lifting**, the wakes of the MRSLs are significantly steered upward. Additionally, the cores of the wakes
(indicated by the light-blue surfaces) are mostly redirected away from the frontal areas of the MRSLs, resulting in much higher
$P^R$ for the downstream MRSLs compared to the case **Without-lifting** (see Figure 9). Furthermore, it is observed that the
wakes' positions are further elevated as the flow progresses deeper into the regenerative wind farm, indicating that the effects
of **Upward-lifting** accumulate progressively across rows. Additionally, arrows on the slices of the velocity contour reveal pairs
of Counter-Rotating Vortices (CRVs) formed by the tip-vortices released by the lifting-devices (these CRVs could be seen
clearer in Figure 14 with the quivers). These CRVs lift the exhaust wakes upward and spread them laterally, simultaneously
bringing down fresh, clean flows from above, thereby replenishing the lower layers, where MRSLs are situated, with higher
energy flows. These CRVs enhance the vertical energy entrainment process by promoting mixing in the vertical direction. See
Sections 4.4 and 4.5 for further discussions on CRVs and the vertical energy entrainment process.

In the case **Downward-lifting**, the wakes of the upstream MRSLs are also steered away from the frontal areas of the
downstream MRSLs, reducing the wake losses experienced by the downstream units. However, the presence of the ground
makes the dynamics of the case **Downward-lifting** quite different from the case **Upward-lifting**. In the **Downward-lifting**
scenario, the wakes are initially directed downward. Then, they are quickly forced to spread sideways as the ground prevents
further downward penetration. As the wakes accumulate on the sides as going deeper into the regenerative wind farm, they
eventually start to move upward. Like the **Upward-lifting** case, CRVs are also present in the **Downward-lifting** case but
rotate in the opposite direction. In this configuration, the CRVs bring fresh, clean flow down from above at the centerlines of
the MRSLs while steering the exhausted wakes downward and sideways.

It is important to note that the purpose of the lifting-devices is not limited to steering the wakes vertically. In fact, the primary
goal of the lifting-devices is to introduce a vertical advection process that enhances vertical mixing, as stronger vertical mixing
leads to stronger vertical energy entrainment. A key aspect of Figure 10 is that the blueish areas in the streamwise velocity





**Figure 10.** Three-dimensional flow structures of the regenerative wind farms around their mid-column. Cases **Without-lifting**, **Upward-lifting**, and **Downward-lifting** are plotted at the top, middle, and bottom, respectively. MRSLs are represented by surfaces in deep blue and deep red, which indicate their rotors and wings. The iso-surfaces in light blue depict the wakes of the MRSLs, corresponding to where $u/u_\mathrm{ref} = 0.65$. Additionally, sections with contours of streamwise velocity in $x$-planes are plotted, with arrows indicating the directions of the in-plane velocity. Note that the arrows' lengths are scaled by the square root of in-plane velocity's norm. The frontal projections of the MRSLs are illustrated with light green squares.



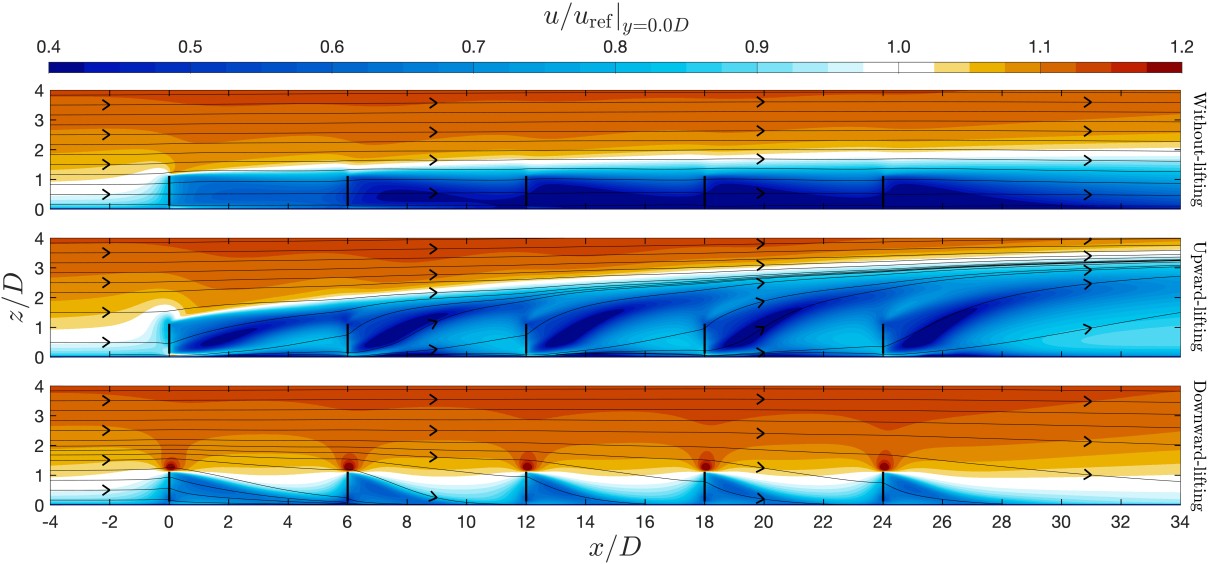

**Figure 11.** Contours of streamwise velocity $u$ of the regenerative wind farms with the cases **Without-lifting**, **Upward-lifting**, and **Downward-lifting** in Table 3. The slices are cut at $y/D = 0.0$ and $u_{\mathrm{ref}} = 10$ m/s. The contours are superimposed with the streamlines based on the in-plane velocity ($u$ and $w$). Thick black lines represent the positions of MRSLs.

contours (areas that $u < u_{\mathrm{ref}}$) for the two cases with lifting-devices are significantly larger than those without, indicating a more pronounced mixing process in the cases with lifting-devices, thus demonstrating their effectiveness.

### 4.3.2  Streamlines

The contours of streamwise velocity $u$, superimposed with streamlines for the three cases in Table 3, are shown in Figure 11. These contours are based on the data on the slices at $y/D = 0.0$, corresponding to the middle of the mid-column. In the

case **Without-lifting**, no significant vertical flows are observed, as indicated by the streamlines, suggesting that the vertical advection process is generally absent. In contrast, for the cases with lifting-devices (**Upward-lifting** and **Downward-lifting**), the streamlines show steep slopes right behind the MRSLs, indicating strong vertical advection and significant vertical mixing.

Additionally, it can be observed that the thickness of the wakes (the blueish area) in the **Without-lifting** case remains nearly constant after the $2^{\mathrm{nd}}$ row (around $1.5D$). In the **Upward-lifting** case, the wake thickness progressively increases as it moves

deeper into the regenerative wind farm, growing from around $1.0D$ to $3.5D$. On the other hand, in the **Downward-lifting** case, the wake thickness decreases with each subsequent row of MRSLs, dropping from $1.0D$ to $0.5D$. However, it should be noted that the surfaces in Figure 11 are confined to $y/D = 0.0D$. If the surfaces were shifted along the $y$-direction, it would be evident that the wakes in both the **Upward-lifting** and **Downward-lifting** cases penetrate higher than in the **Without-lifting** case (thickness of wake for the case **Downward-lifting** can reach to around $2.1D$), as it can be confirmed with the contours of

$u$ displayed in Figure 10. This again demonstrates that lifting-devices enhance vertical mixing within regenerative wind farms.





### 4.3.3 Lateral-averaged streamwise velocity profiles

This subsection explores the lateral-averaged velocity profiles in the regeneratvie wind farms. Two lateral-averaging ranges are considered, which are $-0.5 \leq y/D < 0.5$ and $-2.5 \leq y/D < 2.5$, and their lateral-averaged velocities are denoted as $<u>_{\pm 0.5D}$ and $<u>_{\pm 2.5D}$, respectively. Note that $<u>_{\pm 0.5D}$ averages over the frontal area of MRSLs situated in the mid-column, while $<u>_{\pm 2.5D}$ averages over the entire mid-column. Figure 12 presents the vertical profiles of $<u>_{\pm 0.5D}$ (left) and $<u>_{\pm 2.5D}$ (right) at $x/D = 22.0$, which are located $4D$ downstream from the $4^{\text{th}}$ row of the regenerative wind farms. This position is selected because it is the last row of MRSLs before exiting the regenerative wind farms, and the distance of $4D$ is far enough from the upstream MRSLs, while the induction effects of the downstream MRSLs are minimal.

In the plot of $<u>_{\pm 0.5D}$ profiles (left of Figure 12), it is evident that both case **Upward-lifting** and **Downward-lifting** exhibit significantly larger values for $<u>_{\pm 0.5D}$ around the heights of the MRSLs compared to the case **Without-lifting**, as already reflected in the values of power output reported in Figure 9. Additionally, the shapes of the velocity profiles differ significantly between the two lifting configurations. In the case **Downward-lifting**, the profiles closely resemble the freestream profiles, suggesting that the flow's mean kinetic energy (MKE) is being replenished. Upon closer inspection, between $1.5 < z/D < 3.0$, the $<u>_{\pm 0.5D}$ profiles for the case **Downward-lifting** are slightly higher than those of the freestream. This is related to the strong downward vertical velocities around $y/D = 0.0$, which entrain higher streamwise velocity from the upper layers to the lower ones. In contrast, for the case **Upward-lifting**, the $<u>_{\pm 0.5D}$ profiles decrease with $z$ from $z \simeq 0.2D$ to $z \simeq 1.5D$, which are atypical velocity profiles for standard atmospheric boundary layers. These shapes indicate that the wakes of the MRSLs are channeled upward in case **Upward-lifting** and also indicate that the MKE entrainment is primarily from the sides of MRSLs at the lower layers.

For the profiles of $<u>_{\pm 2.5D}$, notably, the case **Downward-lifting** underperforms the case **Without-lifting** around the height of the MRSLs ($0.12 < z/D < 1.12$). This is mainly because that the MRSLs of the case **Downward-lifting** have extracted more power from the flow at these positions, and the induced drag from the lifting-devices also negatively impacts $<u>_{\pm 2.5D}$. In contrast, the case **Upward-lifting** still significantly outperforms the case **Without-lifting** around the height of the MRSLs, even though it also extracts more energy and introduces induced drag as case **Downward-lifting**. This difference is because the case **Upward-lifting** ejects most of its exhausted wakes upward, while the wakes in the case **Downward-lifting** are mostly trapped at lower altitudes.

For both $<u>_{\pm 0.5D}$ and $<u>_{\pm 2.5D}$ in Figure 12, it is evident that at higher altitudes (larger $z/D$), case **Upward-lifting** has more pronounced effects on altering the velocity profiles compared to case **Downward-lifting**. This observation aligns with the circulation-based analysis carried out in the later section (Section 4.4), where it was found that the positions of CRVs ($z_{\Gamma_x}$) for case **Upward-lifting** progressively rise as the flow moves deeper into the regenerative wind farm, while this is not the case for **Downward-lifting**. Additionally, this observation further suggests that the **Upward-lifting** configuration may have inherent advantages in enhancing vertical entrainment, as it can extend its effects to higher layers of the ABL compared to the **Downward-lifting** configuration.



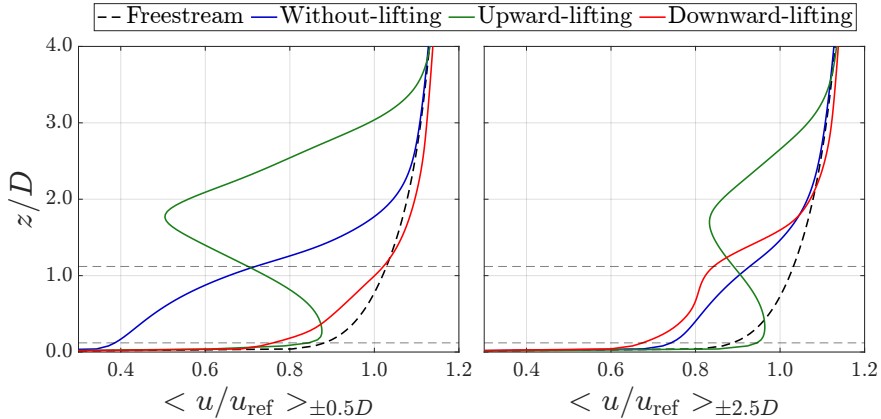

**Figure 12.** Lateral-averaged streamwise velocity profiles. The velocity profiles are sampled at $4D$ after the $4^{\text{th}}$ row of the wind farm ($x/D = 22$). Lateral averaging for the left plot is $-0.5D \leq y < 0.5D$, which covers the frontal area of MRSL of the mid-column. For the right plot, the lateral averaging range is $-2.5D \leq y < 2.5D$, covering the entire mid-column. The freestream profiles are based on the results of the case **Without-lifting** at $x/D = -2D$. **Without-lifting**, **Upward-lifting**, and **Downward-lifting** correspond to the case name in Table 3.

### 4.3.4 Vorticity fields

The fields of streamwise vorticity $\omega_x$ on the selected $x$-planes (the same as those in Figure 10 except for $x/D = -2.0$ is dropped) for cases **Without-lifting**, **Upward-lifting**, and **Downward-lifting** are plotted in Figure 13. In this figure, large-scale vortical structures appear in the wakes of both the **Upward-lifting** and **Downward-lifting** cases, which we call CRVs (counter-rotating vortices). It is evident that CRVs are absent in the wake of the case **Without-lifting**. Additionally, the plots show that CRVs increase in size as they progress deeper into the regenerative wind farms for the cases with lifting-devices.

As described earlier, these CRVs originate from the tip-vortices released by the lifting-devices of the MRSLs (see Figure 2 for three-dimensional representations of CRVs with iso-surfaces of $|\omega_x|$). Thus, as the downstream rows release their tip-vortices, the existing CRVs are strengthened, as can be assessed qualitatively in Figure 13. Furthermore, visual inspection reveals that the centers of the CRVs rise as the flow passes through more rows of MRSLs in the case **Upward-lifting**, while in the case **Downward-lifting**, the centers of the CRVs are observed to be pushed primarily sideways. These observations highlight that

the dynamics of CRVs depend on the lifting configurations of the MRSLs, which is further discussed in Section 4.4.

### 4.4 Quantification of counter-rotating vortices

Utilizing circulation, this section assesses the CRVs (counter-rotating vortices) identified in Figures 10 and 13 in quantified manners. Based on the fields of $\omega_x$, streamwise circulations $\Gamma_x$ of all the seven cases in Table 3 are calculated to represent their CRVs' strengths, which are presented in Figure 14. The values of $\Gamma_x$ in Figure 14 are obtained using Equation 21. Stokes'

theorem is applied in Equation 21, with $C$ being the contour bounding the surface $S$.



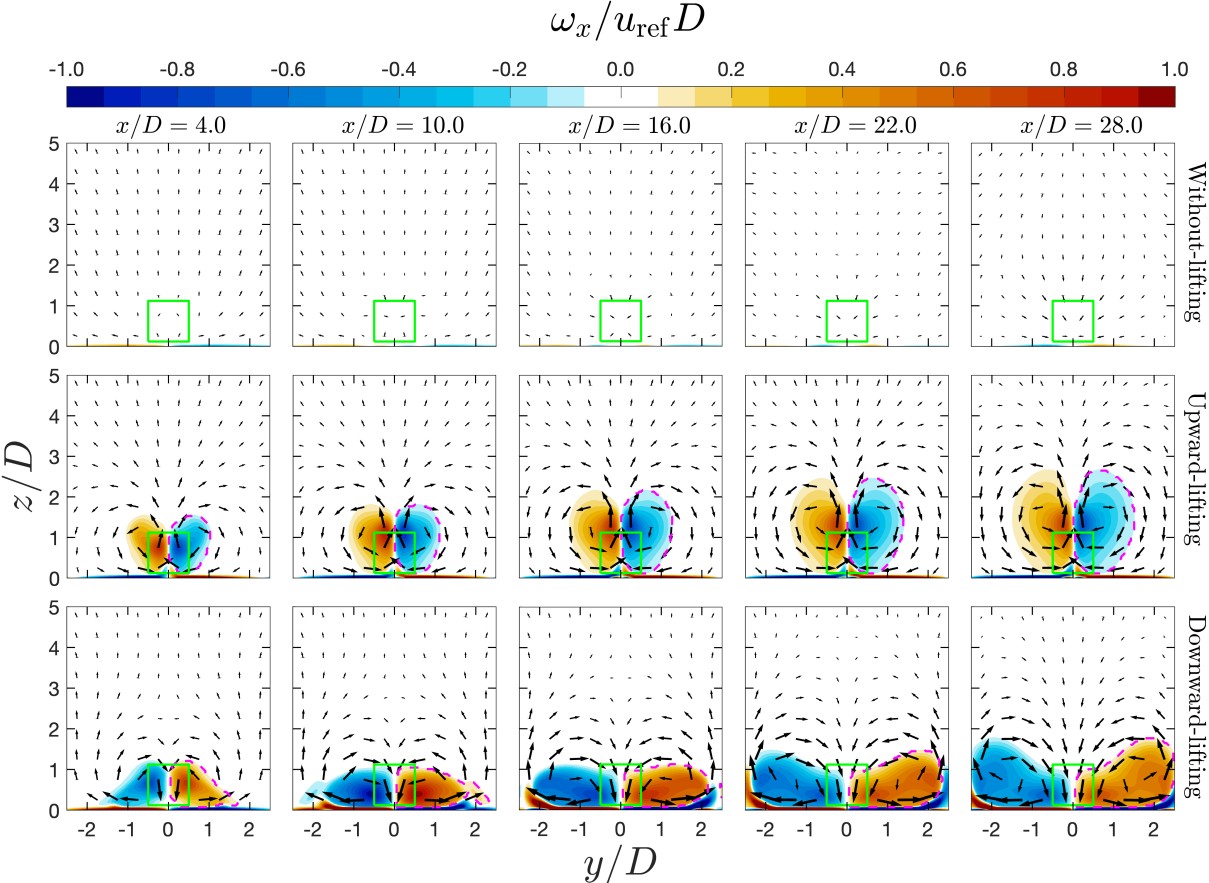

**Figure 13.** Contours of streamwise vorticity $\omega_x$ on $x$-planes at different $x$-positions. The $x$-positions are indicated at the top of each column. These $x$-positions are the same as those sections in Figure 10, except $x/D = -2.0$ is excluded. The cases **Without-lifting**, **Upward-lifting**, and **Downward-lifting** in Table 3 are plotted in the top, middle, and bottom rows, respectively. The frontal projections of the MRSLs are illustrated with light green squares. The vortical structures enclosed by the dashed-magenta-lines are used to calculate the streamwise circulation-related quantities analyzed in Section 4.4. Note that the lengths of the arrows are scaled by the square root of in-plane velocity's norms.



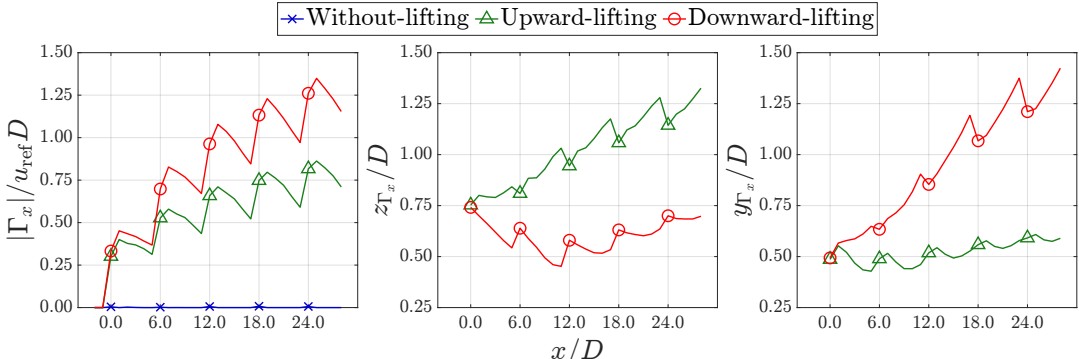

**Figure 14.** Plots of $|\Gamma_x|$ (strength of CRVs, left), $z_{\Gamma_x}$ (heights/$z$-positions of the cores of CRVs, middle), and $y_{\Gamma_x}$ (lateral positions/$y$-positions of the cores of CRVs, right), respectively. They are calculated based on Equations 21 and 22. The example of the considered areas to calculate these quantities are enclosed by the dashed-magenta-lines in Figure 13. See the text for more details. Note that the $x$-ticks are made the same as the $x$-positions of the placed MRSLs.

$$\Gamma_x = \oint_C (u,v,w) \cdot \mathrm{d}\boldsymbol{l} = \int_S \nabla \times (u,v,w) \cdot \mathrm{d}\boldsymbol{A} = \int_S (\omega_x, \omega_y, \omega_z) \cdot \mathrm{d}\boldsymbol{A} = \int_S \omega_x \, \mathrm{d}A \qquad \text{for } \mathrm{d}\boldsymbol{A} \| \hat{\boldsymbol{e}}_x \tag{21}$$

In this work, the strengths of CRVs are defined by the magnitudes of $\Gamma_x$ (denoted as $|\Gamma_x|$) calculated using the right-most of Equation 21. Moreover, the positions of the CRVs are defined based on the *center of gravity (CoG)* of the vortical structure (Saffman, 1995), which is calculated through Equation 22, where $z_{\Gamma_x}$ and $y_{\Gamma_x}$ are defined as the $z$ and $y$-positions of CRVs, respectively. Note that when calculating the $\Gamma_x$ related quantities in the current work, only the regions within $0.0 < y/D < 2.5$ and $0.0 < x/D < 5.0$ are considered. Furthermore, only the $\omega_x$ with the prevailing sign in that region is considered. Examples are illustrated in Figure 13, where $\Gamma_x$ as well as $z_{\Gamma_x}$ and $y_{\Gamma_x}$ are calculated based on the regions enclosed by the dashed-magenta-lines.

$$z_{\Gamma_x} \triangleq \frac{\int_S z \, \omega_x \, \mathrm{d}A}{\int_S \omega_x \, \mathrm{d}A}, \qquad y_{\Gamma_x} \triangleq \frac{\int_S y \, \omega_x \, \mathrm{d}A}{\int_S \omega_x \, \mathrm{d}A} \tag{22}$$

### 4.4.1 Strengths of CRVs

As indicated by the values of $|\Gamma_x|$ in the left of Figure 14, for both the **Upward-lifting** and **Downward-lifting** cases, the strengths of CRVs gradually decrease with larger $x$ before reaching the MRSL of the next row (e.g., $|\Gamma_x|$ drop in the region of $7.0 \leq x/D \leq 11.0$), indicating that CRVs dissipate as they are convected downstream without further perturbation. However, the plot also reveals that the strengths of the CRVs grow stronger as more rows of MRSLs are passed, surpassing the maximum values observed in previous rows. This indicates that the CRVs released by the MRSLs of different rows accumulate, reinforcing their strengths row by row. Stronger CRVs result in stronger vertical flows, making the vertical advection process



more significant. Furthermore, based on $|\Gamma_x|$ in Figure 14, it appears that the strengths of the CRVs have not yet reached their maximum or asymptotic value at the $5^{\text{th}}$ row, suggesting that the strengths of the CRVs may continue to accumulate if the regenerative wind farms have more rows of MRSLs.

### 4.4.2 Positions of CRVs

The positions of the CRVs' cores are quantified using $z_{\Gamma_x}$ and $y_{\Gamma_x}$ introduced earlier. They are plotted in the middle and right of Figure 14. $z_{\Gamma_x}$ and $y_{\Gamma_x}$ indicate the vertical positions (heights) and the lateral positions of the CRVs' cores, respectively.

The self-propelling property of CRVs can be observed by checking the values of $z_{\Gamma_x}$. Specifically, in case **Upward-lifting**, the CRVs' cores rise progressively as they move deeper into the regenerative wind farm. Conversely, in case **Downward-lifting**, the heights of the CRVs' cores gradually decrease starting from the $1^{\text{st}}$ row. However, the positions of $z_{\Gamma_x}$ reach a minimum around the $3^{\text{rd}}$ row of the regenerative wind farm in case **Downward-lifting**, after which they begin to rise. This is primarily due to the presence of the ground and the induced flow from the MRSLs in the side columns (see Figure 2 for a plot illustrating all three columns).

Similarly to $z_{\Gamma_x}$, the $y$-positions (lateral positions) of CRVs, $y_{\Gamma_x}$, also depend on the lifting configurations. In the case **Upward-lifting**, $y_{\Gamma_x}$ consistently remains around $y/D = 0.5$ from the $1^{\text{st}}$ row to the $5^{\text{th}}$ row. Conversely, in the case **Downward-lifting**, $y_{\Gamma_x}$ shifts increasingly outward as the flow travels in the positive $x$-direction. This outward shift of the CRVs is due to the boundary condition imposed by the ground, which can be interpreted through the method of *image vortices* (Saffman, 1995). The presence of the ground also influences the locations of $y_{\Gamma_x}$ in the case **Upward-lifting**, causing them to tend toward $y/D = 0.0$ between any two consecutive rows. However, since $z_{\Gamma_x}$ for the case **Upward-lifting** are located much higher than those in the **Downward-lifting** cases, the effects of the ground are much less significant.

An important aspect to mention is that when the positions of the CRVs' cores are higher (larger $z_{\Gamma_x}$), they may be more capable of entraining flow energy from higher altitudes, which could be beneficial for the vertical entrainment process. Therefore, even when the vertical forcing of **Upward-lifting** and **Downward-lifting** are very similar, the case **Upward-lifting** may offer inherent advantages in this regard.

### 4.5 Control volume analysis on energy budget

In this subsection, the energy transport process of the simulated regenerative wind farms is explored using the control volume approach. The calculations are based on Equation 19, derived in Section 3.5. Six terms are considered, which are *MKE (mean kinetic energy) advection*, *MKE diffusion plus pressure work*, *TKE advection plus diffusion*, *Power extraction (by the MRSL's rotor)*, *TKE dissipation*, and *Residuals*. Five control volumes (CVs) are examined, labeled from A to E (see Figure 6). Each CV encloses an MRSL in the mid-column, covering a range from its $4D$ upstream to its $2D$ downstream. This range is designed to assess the energy sources and sinks of the MRSL in a specific CV. The vertical and lateral ranges are $z_{\text{rc}} - 0.5D \leq z \leq z_{\text{rc}} + 0.5D$ and $-2.5D \leq y \leq 2.5D$, encompassing the entire mid-column. The results of the control volume analysis for the three cases listed in Table 3 are presented in Figure 15.





For the case **Without-lifting** (left of Figure 15), it can be observed that the energy within the CV is primarily supplied
by the advection of MKE and the turbulent shear stress (MKE diffusion plus pressure work), with both terms having similar
magnitudes. Notably, neither of these energy sources alone could supply the power extracted by the MRSL rotors. At first
glance, it may appear that the advection of MKE continues to supply energy to the CVs in the case **Without-lifting**. However,
this is because the wakes of the MRSLs accumulate row by row, continuously depleting the MKE transported in the streamwise
direction (freestream MKE), while the contribution from vertical advection is almost negligible. This interpretation is supported
by the flow fields shown in the top panels of Figures 10 and 11.

For the cases **Upward-lifting** and **Downward-lifting** (middle and right of Figure 15), unlike the case **Without-lifting**, the
contribution of MKE advection is much greater than the work done by turbulent shear stress, with MKE advection alone being
sufficient to support the energy extraction by the MRSLs after the $2^{nd}$ row. Furthermore, the energy supplied to CVs by MKE
advection in these two cases is primarily due to the vertical advection process, driven by the strong vertical velocity component
(see Figure 11). This vertical energy entrainment process differs significantly from conventional wind farms, which mostly rely
on Reynolds shear stress (turbulent shear stress) (Porté-Agel et al., 2020; Calaf et al., 2010; VerHulst and Meneveau, 2014).
Due to the indirect nature of energy entrainment by Reynolds shear stress, its magnitudes are naturally less than that of energy
entrainment by advection. The latter directly injects higher energy flows into the control volumes, while the former relies on a
secondary process involving Reynolds shear stress and the shear layer.

By closely inspecting Figure 15, it can be observed that the MKE advection term is higher for the case **Upward-lifting**
compared to the case **Downward-lifting**. However, the MKE diffusion plus pressure work term (representing the work done by
turbulent shear stress) contributes negatively in the case **Upward-lifting** after the $2^{nd}$ row. This phenomenon can be explained
by the lateral-averaged velocity profiles shown in Figure 12, where the $<u>_{\pm 2.5D}$ profiles for the case **Upward-lifting**
decrease significantly with increasing $z$ in the range of $0.2 < z/D < 1.5D$, causing the shear to impact energy entrainment
negatively. This highlights one of the key differences between **Upward-lifting** and **Downward-lifting**.

## 5   Conclusions

This study conducted numerical investigations of regenerative wind farms. Regenerative wind farm is a newly proposed wind
farm concept that consists of innovative wind harvesting systems, which are the Multi-Rotor System with lifting-devices
(MRSLs, see Figure 1). In these regenerative wind farms, wake recoveries of MRSLs were engineered to be much faster
compared to conventional Horizontal-Axis Wind Turbines (HAWTs), significantly reducing the power losses due to wake
interactions. These enhanced wake recoveries were achieved by altering the vertical entrainment processes. Instead of turbulent
mixing, the entrainment processes were facilitated by the vertical flows induced by tip-vortices generated by the lifting-devices
of MRSLs (see Figure 2). To gain a comprehensive understanding of how the MRSLs' lifting-devices affect the entrainment
processes, cases with different configurations for lifting-devices were tested.

Our results showed that, as the magnitudes of the vertical force were similar to the thrust of MRSLs, the power outputs of
MRSLs with the lifting-devices could be more than tripled compared to those without after the $3^{rd}$ row of the regenerative

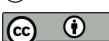


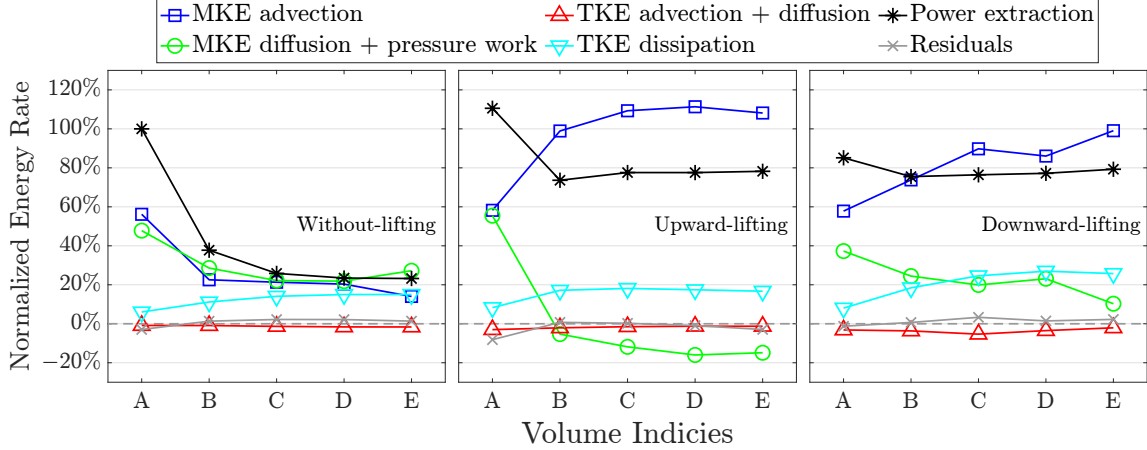

**Figure 15.** Normalized energy transport rates of the terms in Equation 19 based on control volumes. The normalization is done by dividing the values by the rotor power of the MRSL situated at $1^{\text{st}}$-row-mid-column of case **Without-lifting**, which is $P^R\big|_{1^{\text{st}},\text{mid}}^{\mathbf{WL}}$. The cases **Without-lifting**, **Upward-lifting**, and **Downward-lifting** are plotted in the left, middle, and right, respectively. The alphabets at the abscissa refer to the indices of the control volumes (see Figure 6), where volume A is the most upstream one while volume E is the most downstream one. Each volume encloses an MRSL of the mid-column, covering from its $4D$ upstream to its $2D$ downstream.

wind farms (see Figure 4.2), diminishing the wake losses from around $75\%$ to $25\%$. This significant increase in power output highlights the great potential of the regenerative wind farm. Specifically, to deliver the same amount of power, regenerative wind farms would require only half the land area compared to conventional wind farms with HAWTs because they have much

higher wind farm efficiencies (see Table 4). This land use reduction could lower the overall cost of wind energy, making renewable energy more affordable.

Further examinations of how regenerative wind farms could achieve significantly higher power output were conducted by analyzing the flow fields using both qualitative and quantitative methods. Two-dimensional contour plots and three-dimensional iso-surfaces illustrated that the low-velocity wakes of MRSLs were guided vertically upward, while high-velocity fresh flows

were directed downward, replenishing the available power for MRSLs located further downstream. Circulation-based analysis revealed that the strengths of Counter-Rotating Vortices (CRVs), which are the tip-vortices generated by the MRSLs' lifting-devices, accumulated progressively as the flow moved deeper into the regenerative wind farms. These CRVs are responsible for inducing the vertical advection process, with stronger CRVs leading to stronger vertical entrainment processes. Energy budget analysis based on control volumes indicated that wind farms with MRSLs equipped with lifting-devices underwent a

much stronger energy recovery than those with multi-rotor systems lacking such devices. Moreover, the analysis confirmed that the primary contributor to wake recovery in cases with lifting-devices was the vertical advection process, contrasting with conventional wind farms, where wake recovery predominantly relies on turbulent shear (Calaf et al., 2010; VerHulst and Meneveau, 2015). These analyses thoroughly investigated the underlying physics of how regenerative wind farms can achieve significantly higher power outputs.





The results and analysis from this study suggest that the concept of regenerative wind farms could potentially lead to wind farms with much higher farm efficiencies than their conventional counterparts. A series of future research efforts is recommended to fully understand the potential of regenerative wind farms and MRSLs. Several key aspects related to aerodynamics are outlined below. First, conducting simulations using higher-fidelity models, such as large eddy simulations, would be able to better resolve the aerodynamics within the regenerative wind farms. Additionally, investigating whether the stability properties of atmospheric boundary layers influence the dynamics of CRVs would be of significant interest. Moreover, exploring how the layouts of regenerative wind farms and the inflow directions impact their efficiency is also worth pursuing. Furthermore, experimental studies on regenerative wind farms and developing MRSL's prototypes should be considered top priorities, as the ultimate goal is to transform this innovative concept into a real-world application. Certainly, there are numerous other practical challenges beyond aerodynamics, such as the structural integrity of MRSLs, the economic feasibility of regenerative wind farms, and others. These aspects are also critical, and addressing them adequately will be necessary to bring the concept of regenerative wind farms to a commercial stage.

*Code and data availability.* The settings, including the custom library `flyingActuationDiskSource`, of the simulation cases performed in this research are openly available in the 4TU with url being https://data.4tu.nl/private_datasets/aMVl0A0pekcZXksVuKyG53xhJ8H HMcgjeafFRtM8QbA (Li et al., 2024b). All data used in this work are reproducible through executing these cases.

## Appendix A: Sensitivity test of turbulence models

For the steady RANS simulations, all the fluctuating properties are modeled through the turbulence model, and they are modeled differently depending on the model chosen. Thus, model-related uncertainties arise and conclusions obtained by analyzing CFD results might be affected. To ensure that the conclusions obtained in this work are robust and independent of the chosen turbulence model, a handful of simulations are conducted with several mostly used turbulence models. In addition to the already used $k$-$\omega$ SST model (Menter, 1994), configurations of cases **Without-lifting**, **Upward-lifting**, and **Downward-lifting** in Table D1 are tested with realizable $k$-$\varepsilon$ model (Shih et al., 1995) and RNG $k$-$\varepsilon$ model (Yakhot et al., 1992). These are three of the most popular turbulence models for wind energy-related applications, and note that there is currently no unified standard for the optimum RANS turbulence model (Thé and Yu, 2017; Eidi et al., 2021).

The CFD results of cases with different turbulence models are presented in Figure A1. Except for changing the turbulence model, all other parameters remain the same as cases **Without-lifting**, **Upward-lifting**, and **Downward-lifting** listed in Table 3 (inlet conditions of $\varepsilon$ is set to `atmBoundaryLayerInletEpsilon`). It can be seen that similar results are yielded by RNG $k$-$\varepsilon$ model when compared to $k$-$\omega$ SST model. As for the results with the realizable $k$-$\varepsilon$ model, although it shows the effectiveness of the wings of MRSL is less astonishing, significant improvements are still found by comparing the power outputs of cases with and without lifting-devices. Thus, with the results, it can be concluded that even though the selection of the RANS turbulence model may influence the results in the sense of their absolute values, it does not significantly impact the





main conclusion of this work, which is stating that the installation of the lifting-devices (wings) can dramatically improve the power performers of the downstream wind harvesting systems (MRSLs) in regenerative wind farms.

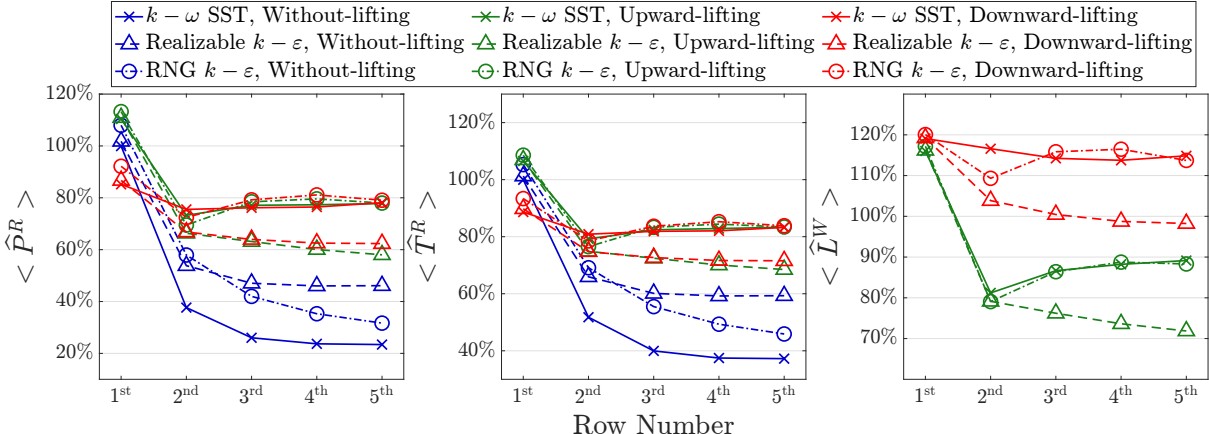

**Figure A1.** Simulations cases with different RANS turbulence models. The configurations used are those of the cases listed in Table 3. $< \widehat{P}^R >$, $< \widehat{T}^R >$, and $< \widehat{L}^W >$ are the normalized row-averaged power, thrust, and vertical force, respectively. The normalization factors are $P^R\big|_{1^{\text{st}},\text{mid}}^{\mathbf{WL}}$ and $T^R\big|_{1^{\text{st}},\text{mid}}^{\mathbf{WL}}$, same as those used in Figures 8 and 9.

## Appendix B: Grid independence test

A grid independence test is carried out to ensure that the discretization error of the CFD simulations does not affect the conclusions drawn. The three cases in Table 3, **Without-lifting**, **Upward-lifting**, and **Downward-lifting**, are tested with three grid sizes $\Delta$. The three tested grid sizes are $\Delta = D/20$, $\Delta = D/25$, and $\Delta = D/30$, and they are labeled as *Coarse*, *Medium*, and *Fine*, and each of them results in a mesh that has 73.9M, 137.8M, and 249.5M cells, respectively. Note that the cases in Table 3 use mesh *Medium*. Also note that, except for adjustment of the grid sizes, all other parameters are kept the same as the cases in Table 3, including the spacings of the actuator elements for wings and the absolute values of the smearing factors ($\varepsilon^R$ and $\varepsilon^W$).

The results of the grid independence test are presented in Figure B1, where $< \Delta P^R >$, $< \Delta T^R >$ and $< \Delta L^W >$ are the relative deviations of $< P^R >$, $< T^R >$, and $< |L^W| >$ from its reference case, respectively. The reference cases are the cases that used mesh *Medium*. The definition of $< \Delta P^R >$ is given in Equation B1, and $< \Delta T^R >$ and $< \Delta L^W >$ are derived in the same way. It can be seen that $< \Delta T^R >$ and $< \Delta L^W >$ of the cases with meshes *Coarse* and *Fine* both fall in the ranges of $\pm 2\%$ for the $1^{\text{st}}$ row and $\pm 4\%$ for all the rows, suggesting that the impacts of grid sizes are minimal for the three considered $\Delta$. For the values of $< \Delta P^R >$ of the **Without-lifting** cases, although they can be up to $6\%$ for the $3^{\text{rd}}$, $4^{\text{th}}$, and $5^{\text{th}}$ rows, their absolute values of the reference $< P^R >$ are relatively small compared to the upstream rows. Due to their relatively





small values, the deviations arising from the upstream rows will be magnified at the downstream. With these results, it can be concluded that the mesh *Medium* ($\Delta = D/25$) is sufficient for the application used in this work.

$$
< \Delta P^R > \text{ of } i\text{th row} \triangleq \frac{< P^R > \text{ of } i\text{th row}- < P^R > \text{ of } i\text{th row with mesh } \textit{Medium}}{< P^R > \text{ of } i\text{th row with mesh } \textit{Medium}} \tag{B1}
$$

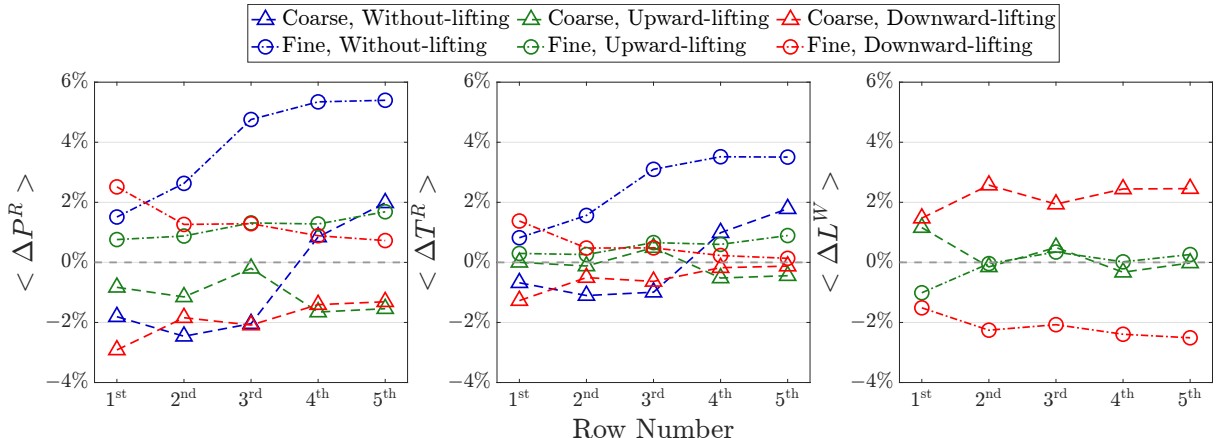

**Figure B1.** Grid Independence test with the three cases listed in Table 3, which are the cases **Without-lifting**, **Upward-lifting**, and **Downward-lifting**. $< \Delta P^R >$, $< \Delta T^R >$ and $< \Delta L^W >$ are the relative deviations of $< P^R >$, $< T^R >$, and $< |L^W| >$ from the cases with mesh *Medium*, where their definitions are in Equation B1.

## Appendix C: Load distribution of MRSLs' wings

The angle of attack and load distributions of the MRSL's wings for cases **Upward-lifting** and **Downward-lifting** in Table 3 are presented in Figures C1 and C2, respectively. Note that the presented angle of attack $\alpha$, streamwise loading $f_x^{\mathrm{AL}}$, and vertical loading $f_z^{\mathrm{AL}}$ are sampled from the mid-column of the wind farms. Definitions of $\alpha$, $f_x^{\mathrm{AL}}$, and $f_z^{\mathrm{AL}}$ are in Equations C1 and C2,
where $\boldsymbol{f}^{\mathrm{AL}}$ is the force exerted by an actuator element of the wings. Focusing on the load distributions of the 1st row MRSL, tip losses can be identified with the $f_z^{\mathrm{AL}}$ profiles. Additionally, with the $f_x^{\mathrm{AL}}$ profiles, it can be seen that induced drags are mainly concentrated around the tips. These results comply with the classical aerodynamics theories (Anderson, 2011), suggesting that the wings' loading predicted by the actuator lines used in this work is reasonable. Some peculiar shapes appear for the loading after the 1st row. This is due to the wakes and vertical flows introduced by upstream MRSLs that complicate the inflow of these
wings. Furthermore, in these two figures, it can be seen that $\alpha$ in the middle of the wings are all $12.5°$, and this $\alpha$ corresponds to $C_l = 2.5$ according to the airfoil polar data of S1223 airfoil (Figure 5), confirming $C_{l,\mathrm{mid}} = 2.5$ holds for these two cases.



$$\alpha \triangleq \begin{cases} -w/u & \text{for the upward lifting cases} \\ w/u & \text{for the downward lifting cases} \end{cases} \tag{C1}$$

$$\boldsymbol{f}^{\mathrm{AL}} = f_x^{\mathrm{AL}} \hat{\boldsymbol{e}}_x + f_z^{\mathrm{AL}} \hat{\boldsymbol{e}}_z \tag{C2}$$

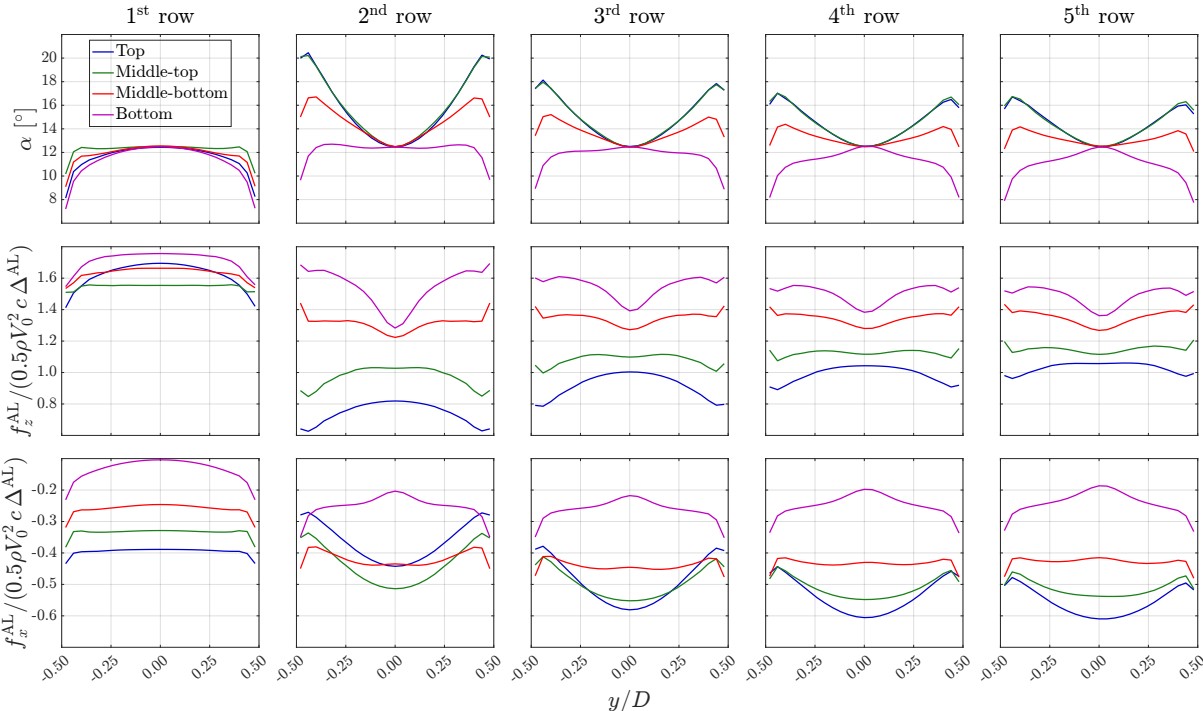

**Figure C1.** The profiles of the angle of attack $\alpha$ (top row), vertical loading $f_z^{\mathrm{AL}}$ (middle row), and streamwise loading $f_x^{\mathrm{AL}}$ (bottom) for the wings of the MRSLs situated in the mid-column for the case **Upward-lifting**.

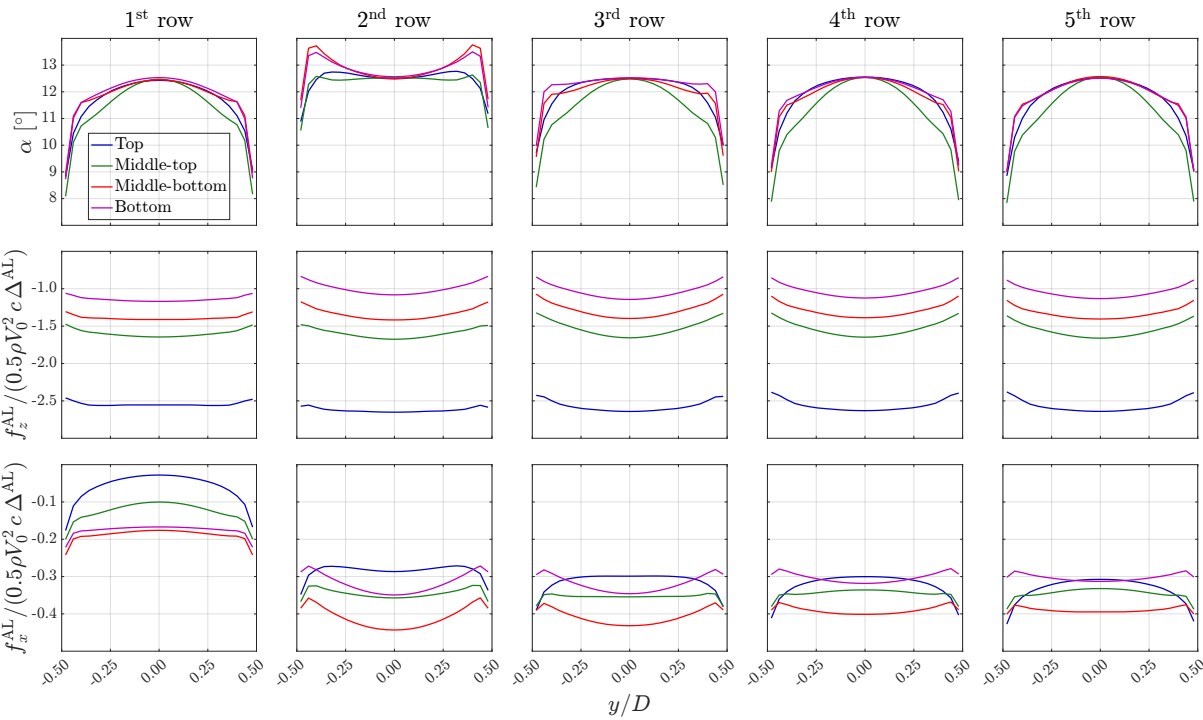

**Figure C2.** The profiles of the angle of attack $\alpha$ (top row), vertical loading $f_z^{\mathrm{AL}}$ (middle row), and streamwise loading $f_x^{\mathrm{AL}}$ (bottom) for the wings of the MRSLs situated in the mid-column for the case **Downward-lifting**.





**Table D1.** The test matrix of the auxiliary cases. These cases have different values of $C_{l,\mathrm{mid}}$, which affects the lift magnitudes of the MRSLs. The values of $C_{l,\mathrm{mid}}$ represent the $C_l$ at the mid-span of the MRSLs' wings of a case (see Section 3.4). Note that the cases marked with an asterisk are the same as those listed in Table 3, where **N0_0**, **U2_5**, and **D2_5** correspond to the **Without-lifting**, **Upward-lifting**, and **Downward-lifting** cases, respectively.

| Case number | Direction of Lift | $C_{l,\mathbf{mid}}$ |
|---|---|---|
| **N0_0*** | - | - |
| **U0_5** | | 0.5 |
| **U1_5** | upward | 1.5 |
| **U2_5*** | | 2.5 |
| **D0_5** | | 0.5 |
| **D1_5** | downward | 1.5 |
| **D2_5*** | | 2.5 |

## Appendix D: Testing MRSLs with different lift magnitudes

To further understand how the magnitudes of MRSL's lift affect the performance of regenerative wind farms, several auxiliary cases are performed. The cases tested are listed in Table D1. Three different lifting magnitudes are tested for each direction of the lift. The lift magnitudes are adjusted by changing $C_{l,\mathrm{mid}}$ (the lift coefficient at the mid-span of the wing) by pitching the wings (see the end of Section 3.4). Both directions of lift are tested with $C_{l,\mathrm{mid}}$ being 0.5, 1.5, and 2.5. Note that the cases marked with an asterisk are the same as those listed in Table 3, where **N0_0**, **U2_5**, and **D2_5** correspond to the **Without-lifting**, **Upward-lifting**, and **Downward-lifting** cases, respectively.

Similarly to Figure 8, Figure D1 presents the normalized row-averaged thrust, lift, and induced drag ($< \widehat{T}^R >$, $< \widehat{L}^W >$, and $< \widehat{D}^W_{\mathrm{ind}} >$) of the MRSLs for the seven cases listed in Table D1. As designed, $< \widehat{L}^W >$ increases with higher $C_{l,\mathrm{mid}}$ values. Additionally, regardless of the lift direction, $< \widehat{T}^R >$ is higher for all the cases with lifting devices compared to the case without. Moreover, from the 2nd row onward, higher $C_{l,\mathrm{mid}}$ correspond to higher $< \widehat{T}^R >$ for both directions of the lift, despite the fact that larger $C_{l,\mathrm{mid}}$ also results in larger $< \widehat{D}^W_{\mathrm{ind}} >$, as shown in the right panel of Figure D1.

Figure D2 summarizes the power performance of the regenerative wind farms for the cases listed in Table D1. It can be seen that the normalized row-averaged power output of the MRSLs ($\widehat{P}^R$) progressively increases with higher values of $C_{l,\mathrm{mid}}$, indicating that the performance of the regenerative wind farms is positively correlated with the lift magnitudes of the MRSLs within the tested range.

## Appendix E: Frandsen wake model

The analytical wake model used in Section 4.2 is known as the Frandsen model, which is proposed by Frandsen et al. (Frandsen et al., 2006) (*region I* is used as the wakes of different columns do not merge). It is derived from momentum analysis over



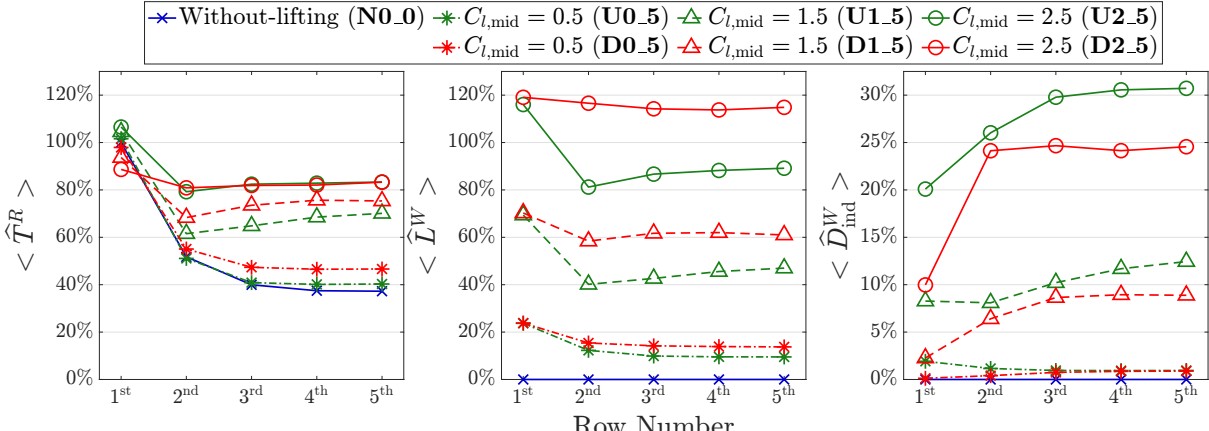

**Figure D1.** The normalized row-averaged thrust of MRSL's rotor ($\widehat{T}^R$, left) together with the vertical ($\widehat{L}^W$, middle) and streamwise ($\widehat{D}^W_{\text{ind}}$, right) force components of the MRSL's lifting-devices. The normalization is done by dividing the reference rotor thrust, which is based on the MRSL at $1^{\text{st}}$-row-mid-column of the case **Without-lifting** (**N0_0**). The legends correspond to the case number introduced in Table D1.

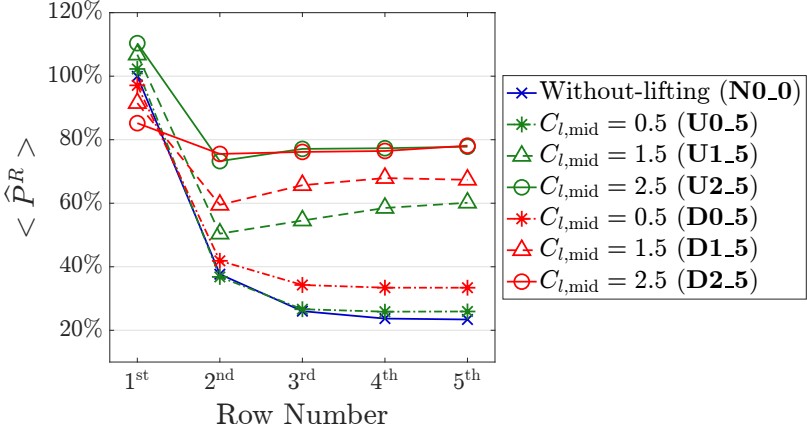

**Figure D2.** The normalized row-averaged rotor power of MRSL's rotor ($\widehat{P}^R$) for the auxiliary cases in Table D1. The normalization is done by dividing the values by the rotor power of the MRSL situated at $1^{\text{st}}$-row-mid-column of the case **Without-lifting** (**N0_0**).

a control volume covering one or multiple wind turbines aligned in the streamwise direction. The inputs of Frandsen model include wind turbine diameter $D_F$, thrust coefficient $C_T$, and the streamwise spacings between the turbines (when there is more than one row of turbines), and the outputs are the wake velocity $u_F$ and wake diameter $D_w$, which vary along the streamwise direction. The equations for the Frandsen model are briefly written in Equations E1 and E2, where $\alpha_F$ is the wake expansion factor that is decided empirically. In current work, $\alpha_F = 0.0629$ is used, based on the CFD results using large eddy simulation reported by Andersen et al. (Andersen et al., 2014).





Note that Frandsen model was developed mainly for horizontal axis wind turbines, thus it is not immediately suitable

for the current work, as MRSL has a square frontal area instead of a circular. Thus, a correction is needed. In this work, $D_F = D_{\mathrm{cir}} = 2D/\sqrt{\pi}$ is used, making $0.25\pi D_F^2 = D^2$ ($D$ is the side length of MRSL). That is, for a circular disk with $D_F$ being diameter, its swept area will be equal to the one for a square MRSL used in this work.

Since in the current work, only the velocity at $x$-positions where there are MRSLs are interested, for simplicity, only the velocity at these positions is calculated. $u_{F,n^{\mathrm{th}}}$ is used to denote the inflow velocity seen by the $n^{\mathrm{th}}$ row of the MRSL predicted

by the Frandsen model, and $x_{n^{\mathrm{th}}}$ is the streamwise position of the $n^{\mathrm{th}}$ row. $D_{w,n^{\mathrm{th}}}$ denotes the wake diameter at $x = x_{n^{\mathrm{th}}}$. For clarity, $u_F$ of the 1st and 2nd rows are explicitly written in Eqaution E3. $u_F$ after the 3rd row are calculated through a recursive method using Equation E4. After having the values of $u_F$ for all the interested rows, relation of Equation E5 is utilized to obtain $P_F^R$ (the power output of the MRSL's rotors predicted by the Frandsen model) based on the one-dimensional momentum theory (Manwell et al., 2010). Note that the values for $C_T$ and $C_P$ are 0.7 and 0.54 as mentioned in Section 2, and the corresponding

power of the 1st row is 29.9MW.

$$D_w(x) = D_F \left( \beta + \frac{\alpha_F\, x}{D_F} \right)^{1/2}, \qquad \beta = \frac{1}{2} \frac{1 + \sqrt{1 - C_T}}{\sqrt{1 - C_T}} \tag{E1}$$

$$u_F = u_{\mathrm{ref}} \left( \frac{1}{2} + \frac{1}{2}\sqrt{1 - 2\left(\frac{D_F}{D_w}\right)^2 C_T} \right) \tag{E2}$$

$$u_{F,1^{\mathrm{st}}} = u_{\mathrm{ref}}, \qquad u_{F,2^{\mathrm{nd}}} = u_{\mathrm{ref}} \left( \frac{1}{2} + \frac{1}{2}\sqrt{1 - 2\left(\frac{D_F}{D_{w,2^{\mathrm{nd}}}}\right)^2 C_T} \right) \tag{E3}$$

$$u_{F,n^{\mathrm{th}}} = u_{\mathrm{ref}} - \left[ \left(\frac{D_{w,(n-1)^{\mathrm{th}}}}{D_{w,n^{\mathrm{th}}}}\right)^2 \left(u_{\mathrm{ref}} - u_{F,(n-1)^{\mathrm{th}}}\right) + \frac{1}{2}\left(\frac{D_F}{D_{w,n^{\mathrm{th}}}}\right)^2 C_T u_{F,(n-1)^{\mathrm{th}}} \right], \qquad \text{For } n \geq 3 \text{ and } x = x_{n^{\mathrm{th}}} \tag{E4}$$

$$P_{F,n^{\mathrm{th}}}^R = 0.5\rho \left(u_{F,n^{\mathrm{th}}}\right)^3 D^2 C_P \propto \left(u_{F,n^{\mathrm{th}}}\right)^3 \tag{E5}$$

*Author contributions.* YL: formal analysis, writing, code development. WY: supervision and technical review. AS: supervision and technical review. CF: conceptualization, supervision and technical review.

*Competing interests.* The contact author has declared that neither they nor their co-authors have any competing interests.



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
