# Peer review of "Numerical Investigation of Regenerative Wind Farms Featuring Enhanced Vertical Energy Entrainment"

_Wind Energy Science, 2024_

## Author Comment (AC1)

**Reply to Reviewers Preprint wes-2024-124**

Title: Numerical Investigation of Regenerative Wind Farms Featuring Enhanced Vertical Energy Entrainment Authors: YuanTso Li, Wei Yu, Andrea Sciacchitano, and Carlos Ferreira

**Authors' reply to comments**

We thank the reviewer for taking the time to read and acknowledging our work. Regarding the interesting inquiries posed by the reviewers, we believe they have all been adequately addressed in this rebuttal as well as in the revised version of the manuscript. We also appreciate the valuable suggestions for further improving the manuscript, and the paper has been revised in light of the comments. The actions taken based on the reviewers' comments are detailed in the following.

**Reviewer#1**

**General comments:**

This paper presents a novel idea which aims to increase the AEP in a wind farm, by forcing the advection of the atmospheric wind into the wake to reenergise the incoming flow for the next wind power generator. This paper presents preliminary RANS simulations using actuator elements to represent the wind turbines.

While I have doubts that this type of system would be largely deployed in the future (due to structural integrity, robustness, reliability and control), the concept is original and it's worth doing this thought experiment.

The paper is globally well written and relatively easy to follow, despite some convoluted turns of phrase. The simulation results are interesting and seem possible to be reproduced as the settings and the code are apparently available.

I have mostly one main concern and one point that troubled me.

**Reply**:**

We sincerely thank the reviewer for acknowledging our work and expressing interest in the concept we have proposed. The inquiries and suggestions raised have been addressed in detail in the following responses.

**Specific comments:**

1. My main concern is the possible large blockage of such a system. When I look at figure 1, I see quite a lot of projected surface compared with more conventional wind turbines. I would imagine that not all the flow would pass through the system and would rather deviate and go around the system. If the flow deflects, it means that there is a smaller mass flow rate through the system, so less energy can be extracted. Could the

authors confirm that in their RANS model, the flow deflection is correctly modelled? For example there are assumptions to define  $u_{ls}^{ele}$  and  $u_{in}^{ele}$ . May this assumption impact the mass flow passing through the system?

**Reply**:**

We appreciate the reviewer for raising this critical aspect concerning the simulation of wind turbine aerodynamics. In our simulation, MRSL is represented using an actuator disk and actuator lines for the rotor and wings, respectively. This approach models the MRSL's presence in the flow field through a body force field, which is known to effectively capture blockage effects, as reflected in our results.

Particularly, in the results of case **Without-Lifting** in Figure 11, the streamwise velocity is observed to slow down before passing through the first row of MRSLs. Furthermore, examining the second bottommost streamline with an arrow, it is clear that it is deflected upward upon encountering the first row of MRSLs, demonstrating the blockage effect noted by the reviewer. Additionally, the thrust predicted by our simulations aligns well with the predictions from one-dimensional momentum theory, indicating that the blockage effects are adequately modeled (note that we used  $C_T^*$  instead of  $C_T$  to model the exerted thrust, see the Section 3.4 for the definition of  $C_T^*$  and the second paragraph of Section 4.1 for the values of the outputted  $C_T$ /thrust). This results further support that the ratio between  $u_{ls}^{ele}$  (streamwise velocity locally sampled at the actuator element) and  $u_{in}^{ele}$  and  $u_{in}^{ele}$  are defined in Section 3.4). Thus, it can be concluded that the axial induction factor ( $a^{ele}$ ) is correctly captured, indicating that the streamwise velocity passing through the actuator disk and, subsequently, the power output predictions are accurate.

However, we acknowledge that the above explanation is valid under the assumption that the effects of the supporting structures are neglected. Incorporating these structures would introduce additional aerodynamic blockage that could impact the power output of MRSL. Accurately modeling such effects would require more complex simulation setups (such as additional actuator elements, the use of an immersed boundary method, or explicitly resolving the full geometry), making the parametric study extremely computational intensive. We have addressed these limitations of the current actuator models in the revised manuscript (see the end of the first paragraph of Section 3.4).

2. The point that troubled me is the terminology with "upward-lifting" and "downward lifting". To me, it seems the upward wind in the wake is due to a downward lift (for example in figure 3, the suction side of the airfoil points downward, so the lift is directed downwards, but it would create an upward wind). This does not impact the results of the paper, but I was doubting if I understood the concept correctly. Could the authors confirm or correct my thoughts and better explain and define this concept in the paper?

**Reply**:**

We thank the reviewer for highlighting that the naming of our cases may cause unnecessary confusion. To clarify, the reviewer's understanding is correct: MRSLs in the case previously referred to as **Upward-lifting** (former name) indeed experience downward lift. We originally named it **Upward-lifting** because the wake of it is being lifted upward.

In light of the doubt posed by the reviewer, we have revised the naming of our simulations to avoid potential ambiguity. Specifically, **Upward-lifting** has been renamed to **Up-Washing** and **Downward-lifting** has been renamed to **Down-Washing**.

*I have a couple of other minor remarks:*

3. Why did the authors choose this airfoil for the lifting devices?

**Reply**:**

S1223 airfoil was selected because it is a representative airfoil profile capable of achieving a high lift coefficient (see Selig et al. 1995 and Selig et al. 1997 in References). Additionally, its moderate camber and thickness make it more practical for real-world implementation. However, we would like to emphasize that the specific airfoil choice is not critical to the performance of the MRSL. The primary objective of the MRSL's wings is to generate strong trailing vortices, which are designed to enhance wake mixing and facilitate wake recovery.

This explanation has been included in the revised manuscript (see the second paragraph of Section 2.2).

4. A Turbulence Intensity of 8% seems fair, but it could be much higher in the reality. As it is mentioned the Turbulent Kinetic Energy plays a minor role, I would interested to know whether the conclusions still hold with a higher TI (such as 20%).

**Reply**:**

The reviewer has highlighted a critical aspect regarding the current deployment of MRSLs. To demonstrate that the concept of regenerative wind farms is robust against variations in inflow turbulence intensity (TI), we conducted additional simulations with different inflow TI values, as detailed in Appendix D. In addition to the previously tested 8%, cases **WL**, **UW**, and **DW** were also tested with inflow TI being 5% and 14%.

The results indicate that while the effectiveness of the MRSLs' lifting devices decreases with higher inflow TI, their performance remains significant. Notably, even at an inflow TI of 14%, which is at or even beyond the upper limit of typical offshore conditions (see Ref. Hansen et al., 2012), the power performance of MRSLs with lifting devices still outperforms those without lifting devices by more than 50%.

We also attempted simulations with an inflow TI of 20%. However, probably because it is unrealistically high for typical offshore environment ( $z_0 = 10^{-4}$  m), the solutions do not converge well, and thus they are not presented.

5. Similarly, the difference of results between the different turbulence models seem quite large. Could the authors precise what could be the reasons for such a large difference between the k-omega and k-epsilon models?

**Reply**:**

We thank the reviewer for raising this very interesting question. Motivated by this, we conducted a brief investigation into the cause of the deviations between turbulence models, which has been documented in Appendix B. In summary, we analyzed the eddy viscosity field  $v_T$ ) predicted by different RANS models and found that the realizable  $k - \varepsilon$  model is significantly more diffusive (predicts higher values for  $v_T$ ) than the other two turbulence models surveyed (the  $k - \omega$  SST and RNG  $k - \varepsilon$  models) for this application.

This increased diffusivity in the realizable  $k - \varepsilon$  model causes the trailing vortices generated by the MRSL's wings to dissipate more quickly, thereby weakening the upwash and downwash effects and slowing the wake recovery rates in the **UW** and **DW** cases. Conversely, the greater diffusivity also promotes the diffusion of mean kinetic energy (MKE), leading to a faster wake recovery rate in the **WL** case compared to results obtained using the other two turbulence models.

**Reviewer#2**

**General comments:**

The article describes a multi-rotor wind energy system with static lifting devices, aimed to increase the momentum entrainment and mitigate wake losses.

The paper is structured well, and the methodology is mostly clearly presented. Perhaps the paper could be shortened by not spelling out every well known concept, for example the RANS equation system with k-omega turbulence model (eqs. 1-4).

**Reply**:**

We thank the reviewer for acknowledging our work and for the valuable suggestion to improve the readability of the manuscript. In response, we have relocated the descriptions of the governing equations for the RANS with the  $k - \omega$  SST model to Appendix A (previously in Section 3.1) and the transport equations of energy to Appendix H (previously in Section 3.5). These adjustments have shortened the main body of the manuscript while ensuring it remains self-contained, as the key equations, specific definitions, and detailed methodologies are still provided in the Appendices.

**Specific comments:**

1. It is not clear how the MRSL appears in the modeling grid. The name indicates several rotors, but Figure 4 indicates one (square) rotor with the diameter D=300m, and 186m hub-height.

**Reply**:**

In our simulations, the MRSL is represented by actuator disks and lines. While the system is conceived to include several sub-rotors, these are not explicitly modeled in our simulations to avoid drastically increasing computational costs and the complexity of parameter studies. In practice, the real-world system would likely resemble the depiction in Figure 1. However, to clarify how the MRSL is represented in the computational domain, Figure 4 is provided. Specifically, the sub-rotors of the MRSL are simplified into a single actuator disk, which is why it appears as a single rotor in Figure 4.

We acknowledge that the previous caption of Figure 4 might not be clear enough and could have caused unnecessary confusion. We have not adjusted the caption clearly stating that Figure 4 is a representation of MRSL in computational domain.

2. Nothing is said about the system integrity and loads on the structure. How does such a system turn into the wind? It could also be assumed that it is fixed and suitable for uni-directional wind climate. In this is the case, please state. While this device appears entirely conceptual and will highly unlikely ever be used at scale, it is however attractive to find an engineering way to capture some of the potential to double the energy density in large wind farms (e.g. Table 4). Have you perhaps tried to add the 5th wing at the lower edge of the system? What about if the wings are installed separate from the multi-rotor structure?

**Reply**:**

We sincerely appreciate the reviewer's thoughtful considerations regarding the MRSL and regenerative wind farm concepts, as well as the valuable suggestions shared with us. Below, we address the points raised:

**About structural integrity and general considerations:**

This study primarily focuses on the aerodynamic aspects of the MRSL as part of demonstrating the regenerative wind farm concept. The primary goal of this manuscript is to provide proof of concept for the MRSL and regenerative wind farms. Consequently,

structural integrity has not been investigated in detail, as it is considered beyond the scope of this work. However, we fully acknowledge that a comprehensive evaluation of structural integrity is critical for real-world implementation. This will be undertaken in future studies if the concepts prove to be feasible, as stated in the final paragraph of the manuscript.

**About yawing:**

We postulate that the MRSL system would yaw against wind direction changes, similar to traditional wind turbines (yawing a large structure of this scale may seem challenging but we think it might be comparable to yawing a 25 MW HAWT as the rotor diameter is expected to exceed 300 m). However, we acknowledge that detailed investigations into yawing mechanisms and their feasibility are necessary before the realization.

**About the fifth wing:**

We did not add a fifth wing at the bottom of the MRSL because wind speeds in that position are relatively low under realistic atmospheric boundary layer (ABL) inflow conditions (as shown in Figure 12). However, this is not a limitation of the MRSL design. A fifth wing could be added if it proves beneficial based on an overall assessment (balancing wake recovery rate, system complexity, and cost).

**About installing the wings and turbines separately:**

We thank the reviewer for this insightful and inspiring question. We believe that installing the wings and turbines separately could achieve similar effects to those demonstrated with the MRSL in this study. An exciting potential of this idea is the possibility of introducing the existing wind farms with wings (i.e., placing wing structures in between the wind turbines), effectively transforming them into regenerative wind farms. This idea holds significant promise and represents a compelling topic for future research. In recognition of this possibility, we have included it as a topic for future exploration in the final paragraph of the Section Conclusions and outlooks.

3. There are many acronyms in the article, but downward-lifting and upward-lifting are for some reason spelled in full over 100 times. Suggest using DL and UL instead.

**Reply**:**

We have changed the case names of **Upward-lifting** and **Downward-lifting** to **Up-Washing** and **Down-Washing**, respectively. And in light of the suggestion given by the reviewer, we have used the acronyms **UW** and **DW** to refer **Up-Washing** and **Down-Washing**, respectively.

**Some other adjustments:**

We have changed the Section title of "Conclusions" to "Conclusions and outlooks", as we think this reflects the contents of the section more precisely.

---

## Author Response (AR2)

**Reply to Reviewers**
**Preprint wes-2024-124-R1**
Title: Numerical Investigation of Regenerative Wind Farms Featuring Enhanced Vertical Energy Entrainment
Authors: YuanTso Li, Wei Yu, Andrea Sciacchitano, and Carlos Ferreira

**Authors' reply to comments**

We sincerely thank the reviewer for taking the time to evaluate our work and for recognizing its contributions. We have carefully considered the valuable suggestions provided and have revised the manuscript accordingly.

Regarding the inquiries posed, while they are indeed important and thought-provoking, some of them fall beyond the scope of the current study. However, to acknowledge their significance, we have explicitly highlighted those aspects in the manuscript as potential directions for future research.

*Reviewer#3*

*General comments*:

*In this work, the authors examined the idea of regenerative wind farms using RANS simulations. Three configurations were tested, i.e., without-lifting, up-washing, and down-washing. Significant increases in power output were demonstrated. It is a very nice work, and the presented results are impressive. Comments are as follows:*

**Reply**:
  We sincerely thank the reviewer for acknowledging our work and expressing interest in the concept we have proposed. The comments given have been addressed in detail in the following responses.

*Specific comments:*

*1. Accurately predicting the induced flow field using RANS methods is challenging. It is nice that the authors tested results from different RANS models. In addition to the contours shown in Figure B.2. It is beneficial to have some quantitative comparisons, such as the time-averaged streamwise, lateral, and vertical velocity profiles.*

**Reply**:
  We appreciate the reviewer's acknowledgment of our efforts in comparing results obtained using different turbulence models. We agree that contour plots in **Figure B2** primarily allow for qualitative comparisons. However, in **Figure B1**, we provide XY-plots illustrating the performance of MRSLs (Multi-Rotor System with Lifting Devices) predicted by employing different turbulence models. These plots offer a quantitative comparison, as the exerted forces and harvested power are directly derived from sampled velocity data, as detailed in **Section 3.4**. Since these output quantities are

closely linked to the flow field parameters, we believe that **Figures B1** and **B2** provide a sufficiently comprehensive assessment of the effects of different turbulence models.

*2. The rotational flow motions caused by the rotor (especially for HAWTs) may also interact with the induced flow fields of the lifting device. In the RANS simulations, such an effect was not taken into account. How will it influence the conclusions of this work?*

**Reply**:

We appreciate the reviewer for highlighting this challenging yet important aspect of MRSL aerodynamics. Indeed, with our current numerical framework, the rotational effects of the sub-rotors in the MRSL are not captured, as detailed in **Section 3.4**. However, we consider this aspect beyond the scope of the present study, and its influence is left for future investigations.

In response to the reviewer's comment, we have now explicitly acknowledged this limitation in the second paragraph of **Section 3.4**: "For instance, the supporting structures of the MRSL are not modeled, the rotational effects of the sub-rotors are not captured, and …". Additionally, we have emphasized this as a potential avenue for future research in **Section 5** (Conclusions and Outlook): "Also, modeling MRSLs with greater detail is desirable, as the effects of their detailed geometry and the rotational of the sub-rotors are omitted in the current work.".

That said, prior experimental work by Broertjes et al. (2024) provides valuable insight. They investigated an MRSL configuration with 16 rotating sub-rotors (vertical-axis wind turbines) and two wings, and their findings align with ours, showing that equipping MRSLs with lifting devices significantly improves wake recovery by enhancing advection. This experimental evidence supports that the secondary effects, including the rotational influence of sub-rotors, do not fundamentally alter the wake recovery mechanism of MRSL. This is now mentioned in the second paragraph of **Section 3.4**: "That said, it is worth noting that Broertjes et al. (2024) have already conducted experiments with an isolated MRSL equipped with rotating sub-rotors and supporting frames, demonstrating that the effectiveness of the lifting devices/wings is not significantly affected by these secondary effects, suggesting omitting them has limited impacts."

We believe these additions help clarify the scope of our study and reinforce the robustness of our conclusions. Moreover, the need for further exploration of secondary aerodynamic effects, including the influence of sub-rotor rotation and supporting structures, are also acknowledged.

*3. Figure 12: it is suggested to add the profiles at several more streamwise positions.*

**Reply**:
We appreciate the reviewer's suggestion. In response, we have added two additional streamwise positions for the lateral-averaged streamwise velocities profiles in **Figure**

**12**. With this adjustment, the evolutions of the lateral-averaged streamwise velocities along the streamwise direction are now more clearly illustrated.

*4. Figure 13: people can only see the relative magnitude of the induced flow field. It is necessary to show quantitative comparisons of the lateral component of the time-averaged velocity.*

**Reply**:

We appreciate the reviewer's valuable suggestion. In response, we have added a dedicated subsection, **Section 4.3.3**, to present and discuss the lateral and vertical velocity components in more detail. This subsection further examines the influence of the released tip vortices (CRVs, counter-rotating vortices) by analyzing the lateral and vertical velocity fields, particularly on how high the flow fields are influenced. Additionally, we now provide quantitative assessments of these velocity components.

To further aid interpretation, we have included absolute scales for the arrows illustrating the in-plane velocity of several plots (**Figures 13**, **14**, **B2**, and **D2**), facilitating a more quantitative evaluation.

We believe these additions enhance the clarity and completeness of our analysis.

*5. It is seen in Figure 10 that the wake becomes larger and is located higher for the up-washing case, while it is split to the sides in the lateral direction for the down-washing case. Eventually, they will together interact with the atmospheric boundary layer. The question is, will the MRSLs remain efficient where there are many rows of them?*

**Reply**:

We appreciate the reviewer for raising this important and practical question. However, the current study focuses on a specific wind farm layout consisting of five rows and three columns, aligned with the freestream direction. The effects of different wind farm layouts and larger farm sizes are beyond the scope of this work and are left for future investigations.

To clarify this limitation, we have explicitly stated in the last paragraph of **Section 4.3.1**: "However, although the significant potential of lifting-devices is presented, their effectiveness in regenerative wind farms with different layouts and sizes remains uncertain, necessitating further investigation in future studies.", and **Section 5** (Conclusions and Outlook): "Furthermore, exploring how the inflow directions, the layouts of regenerative wind farms, and the sizes of the regenerative wind farms impact the farm efficiency is also worth pursuing."

*6. The lifting devices incur additional costs. Will the proposed MRSL still have advantages when considering the levelized cost of energy?*

**Reply**:

We appreciate the reviewer for highlighting this highly relevant and practical concern. However, assessing the levelized cost of energy is beyond the scope of the current study, which focuses primarily on the aerodynamic aspects of MRSLs.

As noted in **Section 5** (Conclusions and Outlook), this question is left for dedicated upcoming investigations: "Certainly, there are numerous other practical challenges beyond aerodynamics, such as the structural integrity of MRSLs, the control strategies and mechanisms of MRSLs (such as yaw control), **the economic feasibility of regenerative wind farms**, and others.